# DISCO: Efficient Diffusion Solver for Large-Scale Combinatorial Optimization Problems

## Abstract

Combinatorial Optimization (CO) problems are fundamentally important in numerous real-world applications across diverse industries, characterized by entailing enormous solution space and demanding time-sensitive response. Despite recent advancements in neural solvers, their limited expressiveness struggles to capture the multi-modal nature of CO landscapes. While some research has shifted towards diffusion models, these models still sample solutions indiscriminately from the entire NP-complete solution space with time-consuming denoising processes, which limit their practicality for large problem scales. We propose **DISCO**, an efficient **DI**ffusion **S**olver for large-scale **C**ombinatorial **O**ptimization problems that excels in both solution quality and inference speed. DISCO's efficacy is twofold: First, it enhances solution quality by constraining the sampling space to a more meaningful domain guided by solution residues, while preserving the multi-modal properties of the output distributions. Second, it accelerates the denoising process through an analytically solvable approach, enabling solution sampling with minimal reverse-time steps and significantly reducing inference time. DISCO delivers strong performance on large-scale Traveling Salesman Problems and challenging Maximal Independent Set benchmarks, with inference duration up to 5.28 times faster than existing diffusion solver alternatives. By incorporating a divide-and-conquer strategy, DISCO can well generalize to solve unseen-scale problem instances off the shelf, even surpassing models specifically trained for those scales.

## 1 Introduction

Combinatorial Optimization (CO) is a fundamental field in both computer science and operations research, encompassing the search for an optimal solution from a finite set of entities. These challenges are widespread in various real-world applications across diverse industries, spanning logistics (Ma et al., 2023; Li et al., 2024), production scheduling (Ye et al., 2024a; Zhang et al., 2024), and resource allocation (Zhao et al., 2021a; 2022). A distinctive characteristic of CO problems is the exponential expansion of their solution space as the problem scale increases. This exponential growth is particularly pronounced in the case of NP-complete (NPC) problems (Garey & Johnson, 1979), representing the most formidable challenges within NP and posing a formidable obstacle to precisely finding an optimal solution within a polynomial time frame.

In recent years, deep learning algorithms have showcased remarkable capabilities in CO problem solving (Choo et al., 2022; Kim et al., 2022). However, these learning-based solvers are susceptible to being misled by the multi-modal landscapes in CO problems (Khalil et al., 2017), wherein the learning agent is required to identify a set of optimal solutions. This multi-modal property complicates the learning, hindering efficient convergence to desired solutions, particularly when confronted with large problem scale (Chen & Tian, 2019; Wu et al., 2021). Diffusion probabilistic models (Ho et al., 2020; Song et al., 2021a) have demonstrated robust capabilities in generation tasks. Of particular interest, Chi et al. (2023) and Huang et al. (2023b) have employed diffusion methods for decision model construction, showcasing their inherent advantages in addressing multi-modal problems. This serves as inspiration for us to explore the application of diffusion methods to CO.

We are not the first to apply diffusion models to CO problems. Graikos et al. (2022) tackle Euclidean Traveling Salesman problems (TSP) by converting each instance into a low-resolution greyscale image and then utilizing a Convolutional Neural Network (CNN) (LeCun et al., 1998) for denoising the solution. Sun & Yang (2023) propose DIFUSCO to explicitly model problem structures with Graph Neural Networks (GNNs) (Gori et al., 2005). Li et al. (2024) further develop DIFUSCO with

an objective-guided, gradient-based search during deployment. Although these approaches show improved performance, they still indiscriminately sample solutions from the entire NPC solution space, simulating a Markov chain for generation with many steps. The incurred time overhead for unproductive solution sampling is a critical bottleneck in applying diffusion solvers to real-world instances, especially when dealing with large problem scales (Xu et al., 2018).

We contend that the potential of diffusion models in addressing large-scale CO problems has yet to be fully discovered. We propose **DISCO**, an efficient **DI**ffusion **S**olver for large-scale **C**ombinatorial **O**ptimization problems. DISCO improves solution quality by restricting the sampling space to a more meaningful domain, guided by solution residues, and enables rapid solution generation with minimal denoising steps. DISCO delivers strong performance on large-scale TSP instances and challenging Maximal Independent Set (MIS) benchmarks, with inference duration up to 5.28 times faster than other diffusion solver alternatives. Through further leveraging the multi-modal property and efficiency of DISCO, we can well generalize it to solve unseen-scale instances with a traditional divide-and-conquer strategy (Fu et al., 2021; Ye et al., 2024b) off the shelf, even outperforming models specifically trained for corresponding scales.

## 2 RELATED WORK

**Combinatorial Optimization** Combinatorial optimization (CO) problems have garnered considerable attention over the years due to their extensive applicability across diverse domains such as logistics (Bello et al., 2016; Kool et al., 2019), production scheduling (Ye et al., 2024a; Zhang et al., 2024), and resource allocation (Zhao et al., 2021a; 2022). However, the exponential growth of the solution space, as the problem scale escalates for these NPC problems (Garey & Johnson, 1979), poses a formidable challenge for finding an optimal solution within a polynomial time frame. Traditional solvers for CO problems can be classified into exact algorithms, approximation algorithms, and heuristic methods. Exact algorithms (Lawler & Wood, 1966; Schrijver et al., 2003), such as dynamic programming (Cormen et al., 2022) and cutting-plane methods (Wolsey & Nemhauser, 2014), aim to exactly find the optimal solution for each test instance. However, they only suit small to medium-sized problems due to the inherent heavy computational complexity. Approximation (Hochba, 1997; Vazirani, 2001) and heuristic (Glover & Kochenberger, 2006; Michalewicz & Fogel, 2013) methods, on the other hand, are used when the problem scale is large or time constraints exist for finding solutions. These methods can find solutions within an acceptable time cost. However, they typically heavily rely on expert knowledge (Helsgaun, 2017; Taillard & Helsgaun, 2019) and cannot guarantee the high quality of the final discoveries.

**Learning for Combinatorial Optimization** With the blossoming of deep learning mechanisms that do not heavily rely on expert knowledge and can be easily adapted to various automated search processes, researchers have widely explored neural solvers for CO problems. These approaches encompass both supervised learning (SL) (Vinyals et al., 2015) and reinforcement learning (RL) (Mnih et al., 2015). From a practical perspective, the choice between SL and RL depends on the availability of problem data. For online operation problems (Seiden, 2002; Borodin & El-Yaniv, 2005), the input data is progressively revealed, and decisions must be made immediately upon data arrival. Such problems require algorithms to make decisions without fully understanding the problem, typically modeled as Markov Decision processes and solved through trial-and-error methods using RL (Zhao et al., 2021b; 2023). Conversely, for offline problems (Papadimitriou & Steiglitz, 1998), all input data and all constraints are fully provided before solving the problems. The decision-makers can fully utilize all relevant information for comprehensive analysis and iteratively improve solution quality. Providing an initial solution by SL and further refining it by decoding strategies (Croes, 1958; Kool et al., 2019; Graikos et al., 2022) has become a common practice (Deudon et al., 2018). Most CO problems can be modeled as decision problems on graphs (Yolcu & Póczos, 2019; Li & Si, 2022; Zhang et al., 2024). Notably, TSP (Bi et al., 2022) and MIS (Darvariu et al., 2021) stand out as two foundations regarding edge and node decision problems. DISCO leverages anisotropic GNNs (Bresson & Laurent, 2018; Joshi et al., 2022) as the backbone to produce embeddings for both graph edges and nodes, adequately demonstrating its superiority on both large-scale TSP and MIS instances through extensive evaluations.

**Diffusion Probabilistic Model** Diffusion probabilistic models (DPMs) (Ho et al., 2020; Song et al., 2021a) are primarily utilized for high-quality generation and have exhibited robust capabilities in generating images (Huang et al., 2023c), audios (Luo et al., 2024), and videos (Ho et al., 2022). This

impressive method was initially formulated by Sohl-Dickstein et al. (2015) and further extended by Ho et al. (2020) through the proposal of a general generation framework. Its principle involves simulating a forward process of gradually introducing noise, followed by training a reverse noise removal model to generate data. These models can further adjust the conditional variables (Dhariwal & Nichol, 2021) during the reverse process to generate data samples that satisfy specific attributes or conditions. In comparison to other generative models such as Generative Adversarial Networks (Goodfellow et al., 2014; Radford et al., 2015), diffusion models demonstrate higher stability during training. This is attributed to their avoidance of adversarial training and they gradually approach the true distribution of the data by learning to remove noise.

**Diffusion for Combinatorial Optimization**  In addition to stable and high-quality generation, DPMs have exhibited a promising prospect for generating a wide variety of distributions (Huang et al., 2023b). This multi-modal property particularly benefits CO problem solving, where multiple optimal solutions may exist and confront the limited expressiveness of previous neural solvers (Gu et al., 2018; Li et al., 2018). Some attempts have been made. Graikos et al. (2022) convert TSP instances into low-resolution greyscale images encoded by CNN. Sun & Yang (2023) propose DIFUSCO to incorporate GNN for problem representation while Li et al. (2024) further develop DIFUSCO with an objective-guided, gradient-based search during deployment. These efforts overlook the inefficient solution sampling from enormous NPC solution space and the slow reverse process of diffusion models, which significantly hampers their practicality for large-scale real-world applications (Xu et al., 2018). DISCO differentiates itself by developing a specialized diffusion process tailored for CO, optimizing both forward and reverse processes. Specifically, DISCO employs an analytical denoising process (Huang et al., 2023a) to quickly produce high-quality solutions with very few denoising steps, while reducing solution space associated with NPC problems by introducing solution residues (Liu et al., 2024). This enhanced efficiency on both solution quality and inference speed further amplifies DISCO's advantages in generalizing to the CO challenge of unseen scales.

## 3 PRELIMINARY

Combinatorial optimization can generically be framed as the task of finding a valid solution $\mathbf{X}_s$ from a discrete solution space $\mathcal{X}_s = \{0, 1\}^N$ for a given instance $s$, while minimizing the task-specific cost function $\text{cost}(\mathbf{X}_s)$ (Papadimitriou & Steiglitz, 1998). The optimal solution $\mathbf{X}_s^*$ is defined as:

$$\underset{\mathbf{X}_s \in \mathcal{X}_s}{\arg\min} \, \text{cost}(\mathbf{X}_s). \tag{1}$$

Taking TSP instances as an example, $N$ represents the edge number, $X_i \in \mathbf{X}_s$ indicates whether the $i$-th edge is selected, and $\text{cost}_s(\mathbf{X})$ means the tour length of $\mathbf{X}$. Parameterized solvers, denoted as $p(\cdot|s)$, are trained to predict the probability distribution over each problem variable. Either supervised learning (Vinyals et al., 2015; Sun & Yang, 2023) or reinforcement learning (Bello et al., 2016; Kool et al., 2019) mechanisms have been extensively explored.

While previous neural CO solvers have shown promising results, they usually suffer from the expressiveness limitation when confronted with multiple optimal solutions for the same graph (Khalil et al., 2017; Gu et al., 2018). Thanks to recent advances in generative models, DPMs have exhibited promising prospects for generating a wide variety of distributions (Ho et al., 2020; Huang et al., 2023b) suitable for CO solving.

DPMs view the input-to-noise process as a parameterized Markov chain that gradually adds noise to the original data $\mathbf{x}_0$ until the signal is completely corrupted, this forward process is first formulated by Sohl-Dickstein et al. (2015) with the definition:

$$q(\mathbf{x}_t \mid \mathbf{x}_0) = \mathcal{N}\left(\mathbf{x}_t; \alpha_t \mathbf{x}_0, \beta_t^2 \mathbf{I}\right), \tag{2}$$

where $\alpha_t$ and $\beta_t$ are the differentiable functions of time $t$ with bounded derivatives, $\mathbf{x}_t$ is noisy data, and $\mathbf{I}$ is the identity matrix. Song et al. (2021b) give proof that this Markov chain can be represented by the following stochastic differential equation:

$$d\mathbf{x}_t = h_t \mathbf{x}_t \, dt + g(t) \, d\mathbf{w}_t, \quad \mathbf{x}_0 \sim q(\mathbf{x}_0), \tag{3}$$

where $h_t = \frac{\mathrm{d} \log \alpha_t}{\mathrm{d}t}$, $g_t^2 = \frac{\mathrm{d}\beta_t^2}{\mathrm{d}t} - 2h_t \beta_t^2$, and $\mathbf{w}_t$ denotes the standard Wiener process (Einstein, 1905).

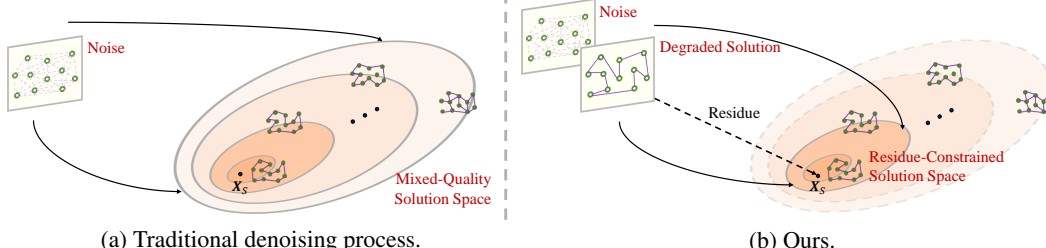

(a) Traditional denoising process.        (b) Ours.

Figure 1: In comparing DISCO's solution sampling with traditional diffusion methods. We define darker colors to represent higher solution quality. (a) Traditional diffusion generation indiscriminately spans the entire mixed-quality solution space, i.e., a significant proportion of the samples do not satisfy problem constraints. (b) DISCO constrains the generations close to the high-quality label $\mathbf{X}_s$ by introducing residues, resulting in a more meaningful, yet smaller, solution space, while preserving the multi-modal properties of the output distributions.

# 4 METHOD

At the outset, we introduce solution residues in Sec. 4.1, which restricts the sampling space for large-scale CO problems to a more meaningful domain, ensuring solution effectiveness while preserving diversity. In Sec. 4.2, we present our analytical denoising process to generate high-quality solutions with minimal reverse-time steps. In Sec. 4.3, we further leverage the multi-modal property and efficiency of DISCO to generalize it to unseen scales with a traditional divide-and-conquer strategy.

## 4.1 RESIDUE-CONSTRAINED SOLUTION GENERATION

The NPC solution space grows exponentially with CO problem scales. The reverse generation covering such an enormous space is inefficient since many samples do not even adhere to problem constraints, as depicted in Fig. 1 (a). We propose our DISCO method to restrict the sampling from the entire NPC space to a more meaningful domain while still preserving the multi-modal property of output distributions. We achieve this by introducing solution residues (Liu et al., 2024) to prioritize certainty besides noises to emphasize diversity, as shown in Fig. 1 (b). The reversed process starts from both noise and an exceedingly economical degraded solution, confining the generated samples close to the high-quality input data. Driving a high-quality solution from a degraded or heuristic one has been verified as effective and is widely adopted in solving various CO problems (Zhao et al., 2022; Zhang et al., 2024). Conditional guidance can decrease the unconditional likelihood of the sample while increasing the conditional likelihood, leading to higher sample quality (Ho & Salimans, 2022).

Given problem instance $s$, parameterized DPM $p(\cdot|s)$ generates conditionally independent probability distribution $\mathbf{x}_0$ for each problem variable, also known as heatmap scores (Fu et al., 2021; Sun & Yang, 2023). Subsequently, task-specific decoding processes (Croes, 1958; Kool et al., 2019) are employed to transform predicted $\mathbf{x}_0$ into discrete solution $\mathbf{X}_s$. We denote $\mathbf{X}_d$ a readily obtainable degraded solution that satisfies problem constraints and solution residues $\mathbf{x}_{res} = \mathbf{X}_d - \mathbf{x}_0$. Take TSP as an example, $\mathbf{X}_d$ can be obtained by connecting vertices in the graph in a sequential order to form a tour. By introducing the residue $x_{res}$, the forward diffusion process is the mapping from the high-quality solution to the mixture of noise and degraded solution:

$$\mathbf{x}_t = \mathbf{x}_0 + (1 - \alpha_t)\mathbf{x}_{res} + \beta_t \boldsymbol{\epsilon}, \quad \boldsymbol{\epsilon} \sim \mathcal{N}(\boldsymbol{\epsilon}; \mathbf{0}, \mathbf{I}), \tag{4}$$

where $\mathbf{x}_0 = \mathbf{X}_s$ denotes the high-quality solution label and $\mathbf{x}_t$ is the noisy solution. According to the reversed process of DDPM (Ho et al., 2020), we can derive the transition probability of the reversed process that is defined as:

$$q(\mathbf{x}_{t-1}|\mathbf{x}_t, \mathbf{x}_0) \propto \exp\left\{-\frac{(\mathbf{x}_{t-1} - \mathbf{u})^2}{2\sigma^2 \mathbf{I}}\right\},$$

$$\mathbf{u} = \frac{\alpha_{t-1}\alpha_t\beta_{t-1}^2 + \alpha_{t-1}^2 - \alpha_t^2}{\alpha_{t-1}\beta_t^2}\mathbf{x}_t + \frac{(\alpha_t^2 - \alpha_{t-1}^2)(1 - \alpha_t)}{\alpha_{t-1}\beta_t^2}\mathbf{x}_{res} + \frac{\alpha_t^2 - \alpha_{t-1}^2}{\alpha_{t-1}\beta_t}\boldsymbol{\epsilon}, \tag{5}$$

$$\sigma^2 = \frac{(\alpha_{t-1}^2 - \alpha_t^2)\beta_{t-1}^2}{\alpha_{t-1}^2\beta_t^2}.$$

The residue prioritizes certainty while the noise emphasizes diversity, so that the solution space for sampling is effectively constrained. For the learning process, the diffusion model only needs to learn the residue between the high-quality label $\mathbf{X}_s$ and the proposed degraded solutions $\mathbf{X}_d$ rather than the original $\mathbf{X}_s$, which simplifies the learning. For the inference process, the generations are confined close to the high-quality label $\mathbf{X}_s$ by introducing residues, allowing the model to efficiently find high-quality solutions while leveraging this meaningful diversity to further improvement.

### 4.2 ANALYTICALLY SOLVABLE DENOISING PROCESS

The residue-constrained denoising process allows for the efficient generation of high-quality solutions; however, typical DDPM usually takes 900~1000 sampling steps for the inference. The slow solving speed significantly limits the practical application of diffusion solvers in real-world CO problems, particularly considering many time-sensitive demands, such as on-call routing (Ghiani et al., 2003) and on-demand hailing service (Xu et al., 2018), not to mention the large-scale operation challenges.

To avoid time-consuming numerical integration and generate high-quality solutions with fewer steps, we substitute the numerical integration process with an analytically solvable form. Inspired by decoupled diffusion models (DDMs) (Huang et al., 2023a), the original mapping in Eq. 4 can be decoupled into an analytical high-quality solution to degraded solution and a zero-to-noise mapping:

$$\mathbf{x}_t = \mathbf{x}_0 + \int_0^t \mathbf{x}_{res} dt + \sqrt{t}\boldsymbol{\epsilon}, \quad \boldsymbol{\epsilon} \sim \mathcal{N}(\boldsymbol{\epsilon}; \mathbf{0}, \mathbf{I}), \tag{6}$$

where $\mathbf{x}_0 + \int_0^t \mathbf{x}_{res} dt$ represents the solution to degradation, and $\sqrt{t}\boldsymbol{\epsilon}$ denotes the zero-to-noise process. More importantly, since there is an analytical solution-to-degradation in the forward process, we can derive the corresponding reversed process with a similar analytical form. In this way, the efficiency of the reversed process can be improved by much fewer evaluation steps, e.g., inference with 1 or 2 steps. More specifically, we employ continuous-time Markov chain with the smallest time step $\Delta t \to 0^+$ and use conditional distribution $q(\mathbf{x}_{t-\Delta t} \mid \mathbf{x}_t, \mathbf{x}_0)$ to approximate $q(\mathbf{x}_{t-\Delta t} \mid \mathbf{x}_t)$, which is formulated by:

$$q(\mathbf{x}_{t-\Delta t}|\mathbf{x}_t, \mathbf{x}_0) \propto \exp\left\{-\frac{(\mathbf{x}_{t-\Delta t} - \mathbf{u})^2}{2\sigma^2 \mathbf{I}}\right\},$$

$$\mathbf{u} = \mathbf{x}_t - \int_{t-\Delta t}^t \mathbf{x}_{res} dt - \Delta t \boldsymbol{\epsilon}/\sqrt{t}, \quad \sigma^2 = \Delta t(t - \Delta t)/t, \tag{7}$$

where $\boldsymbol{\epsilon} \sim \mathcal{N}(\mathbf{0}, \mathbf{I})$. Benefiting from the analytical solution to degradation, we avoid the numerical integration-based denoising and instead directly sample heatmap $\mathbf{x}_0$ with an arbitrary step size, which significantly reduces the inference time.

We provide a theoretical analysis of the equivalence between DISCO and DDM in App. A, supporting the effectiveness of our method. It is important to note that DDMs are not directly adopted by DISCO. We integrate our residue-constrained design with DDMs, leading to refined diffusion processes and training objectives. DISCO can efficiently achieve high-quality solutions by sampling from the constrained solution space with fewer denoising steps, meeting the requirements of large-scale CO.

**Training** We adopt anisotropic GNNs (Bresson & Laurent, 2018; Joshi et al., 2022) as the network architecture of DISCO. Unlike typical GNNs such as GCN (Kipf & Welling, 2016) or GAT (Velickovic et al., 2017) designed for node-only embedding, anisotropic GNNs produce embeddings for both nodes and edges, which are then fed into the diffusion model to generate heatmaps. Practically, we input the noisy solution $\mathbf{x}_t$, the nodes and edges of $\mathbf{X}_d$, and the time $t$ into the anisotropic GNN with parameter $\boldsymbol{\theta}$, predicting the parameterized residue $\mathbf{x}_{res}^{\boldsymbol{\theta}}$ and noise $\boldsymbol{\epsilon}^{\boldsymbol{\theta}}$ simultaneously. Specific implementation details are provided in App. G.

We focus on offline CO problems. Therefore, we train DISCO in an efficient and stable supervised mechanism to discover common patterns from high-quality solutions available for each instance. This also helps circumvent the challenges associated with scaling up and the latency in the inference that arises from the sparse rewards and sample efficiency issues when learning in an RL framework (Ma et al., 2021; Wu et al., 2021), especially at large scales. The training objective is defined as:

$$\min_{\boldsymbol{\theta}} \mathbb{E}_{q(\mathbf{X}_s)} \mathbb{E}_{q(\boldsymbol{\epsilon})} \left[\|\mathbf{x}_{res}^{\boldsymbol{\theta}} - \mathbf{x}_{res}\|^2 + \|\boldsymbol{\epsilon}^{\boldsymbol{\theta}} - \boldsymbol{\epsilon}\|^2\right]. \tag{8}$$

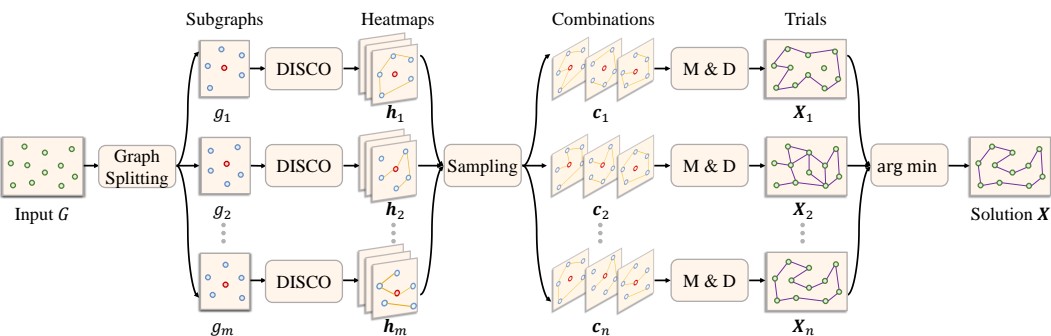

Figure 2: Our multi-modal graph search method, illustrated using a TSP instance for simplicity. M & D denotes merging a combination of heatmaps $\mathbf{c}$ to global heatmap $\mathbf{H}$ and decoding a trial $\mathbf{X}$ from $\mathbf{H}$.

Once trained, the model can be applied to generate heatmaps for a virtually unlimited number of unseen graphs during deployment. These heatmaps are fed into decoding strategies like Greedy (Graikos et al., 2022), Sample (Kool et al., 2019), 2-opt (Croes, 1958), to achieve the final solution.

**Sampling**  Eq. 6 shows the endpoint of the forward process is the mixture of the degraded solution and noise, therefore, we start from the mixture in the sampling process. Given a degraded solution $\mathbf{X}_d$ and a noise $\boldsymbol{\epsilon}$ sampled from the normal distribution, we set $\mathbf{x}_1 = \mathbf{X}_d + \boldsymbol{\epsilon}$ ($t = 1$). Sampling from $\mathbf{x}_1$ instead of $\boldsymbol{\epsilon}$ constrains the sample space from the entire noise domain into a smaller one, ensuring an effective solution. For a $K$-step sampling, we set the step size of each sampling to $1/K$. At each sampling step, we utilize the anisotropic GNN to predict the estimated residue $\mathbf{x}_{res}^{\boldsymbol{\theta}}$ and noise $\boldsymbol{\epsilon}^{\boldsymbol{\theta}}$. In this way, we can solve the reversed process via Eq. 7 iteratively, obtaining the high-quality solution $\mathbf{x}_0$ until $t = 0$. After we sample a probability distribution $\mathbf{x}_0$ from $p_{\boldsymbol{\theta}}(s)$ for instance $s$, we adopt the same operation as Sun & Yang (2023) to obtain the normalized heatmap score $\mathbf{h} = 0.5(\mathbf{x}_0 + 1)$.

### 4.3 Multi-Modal Graph Search

Parameterized solvers trained on specific scales often struggle to generalize well to test instances of different scales (Fu et al., 2021). Training a model from scratch on the target scale or fine-tuning the model includes additional training time within the decision loop, making it impractical for real-world applications that demand an off-the-shelf response. We aim to develop the generalization ability of DISCO to unseen-scale instances. DISCO demonstrates efficiency advantages in both solution quality and inference speed. Its multi-modal output can be further leveraged to enhance solution diversity through a novel divide-and-conquer approach. In contrast, traditional divide-and-conquer strategies (Fu et al., 2021; Ye et al., 2024b) can only produce a single deterministic solution for each sub-problem. Increasing the solution diversity broadens the exploration of the solution space, decreasing the likelihood of getting stuck in sub-optimal solutions (Zhang et al., 2015) and improving generalization performance.

Specifically, we leverage a model $p_{\boldsymbol{\theta}}$ trained on a smaller scale as a base to construct heatmaps for sub-problems $\mathbf{g}$, which are decomposed from the original graph $G$. The scale of sub-problem $g \in \mathbf{g}$ is fixed and close to the training scale of $p_{\boldsymbol{\theta}}$, thereby we can better solve $g$ and smoothly generalize $p_{\boldsymbol{\theta}}$ to the original scale. A detailed pipeline is provided in Fig. 2. For sub-problem decomposition, we adopt a vector $\mathbf{o}_v$ recording the occurrence number of each node of $G$ in all existing subgraphs. In each iteration, we choose the node with the index $\arg\min(\mathbf{o}_v)$ as the cluster center and select the remaining nodes with the k-nearest neighbor rule (Cover & Hart, 1967), forming a subgraph $g$. This process continues until $\min(\mathbf{o}_v)$ exceeds a certain threshold $\omega$. For each $g \in \mathbf{g}$, we resize it to a uniform size.

---

**Algorithm 1** Multi-Modal Graph Search

**Input:** A graph problem $G$ to be solved
**Process:**
1: Pre-train DISCO model $p_{\boldsymbol{\theta}}$ on a small scale
2: Split $G$ into a set of subgraphs $\mathbf{g}$
3: **for** $g \in \mathbf{g}$ **do**
4:     Sample heatmap set $\mathcal{H}$ from $p_{\boldsymbol{\theta}}(g)$ with $q$ different noise $\mathbf{x}_t$
5: **end for**
6: Initialize trial set $\mathcal{X} = \emptyset$
7: **for** $k = 1, 2, 3, \ldots, n$ **do**
8:     Sample $\mathbf{h}$ from each $\mathcal{H}$ as combination $\mathbf{c}_k$
9:     Merge $\mathbf{c}_k$ as a global heatmap $\mathbf{H}_k$
10:     Decode trial $\mathbf{X}_k$ from $\mathbf{H}_k$, add it to $\mathcal{X}$
11: **end for**
12: Select a final trial with $\arg\min_{\mathbf{X} \in \mathcal{X}} \text{cost}(\mathbf{X})$

Leveraging trained DISCO model $p_\theta$, we can generate sub-heatmap score $\mathbf{h}$ for $g$. The solution for the original graph $G$ is obtained by merging all sub-heatmaps, which jointly cover $G$ at least $\omega$ times, through mean aggregation. The merged global heatmap $\mathbf{H}$ is:

$$\mathbf{H}_{ij} = \frac{1}{o_{ij}} \times \sum_{l=1}^{|\mathbf{g}|} \phi(\mathbf{h}_l, i, j), \tag{9}$$

where $\phi(\mathbf{h}_l, i, j)$ represents the heatmap value contributed by $\mathbf{h}_l$ corresponding to index $ij$ of $\mathbf{H}$, with $\phi(\mathbf{h}_l, i, j) = 0$ if no correspondence. The scalar $o_{ij}$ records the occurrence count of edge $ij$ across all subgraphs. Subsequently, we decode the merged heatmap $\mathbf{H}$ into the final solution $\mathbf{X}$. A comparison of various graph merging methods, along with evidence that our graph splitting method helps avoid local optima, is also provided in App. H.

We leverage the multi-modal output of DISCO to enhance the solution diversity and avoid the final solution from getting stuck in the sub-optimum. For each subgraph $g \in \mathbf{g}$, we repeatedly sample a set of heatmaps $\mathcal{H}$ with $q$ different noise $\mathbf{x}_t$. We randomly sample one heatmap $\mathbf{h}$ from each $\mathcal{H}$, combining as a set $\mathbf{c}$ with $|\mathbf{c}| = |\mathbf{g}|$, merging as a global heatmap $\mathbf{H}$, and decoding a solution trial $\mathbf{X}$ from $\mathbf{H}$. This sample process is repeated $n$ times, generating multiple trials as $\mathcal{X}$. We decide the final solution with the minimum cost from $\mathcal{X}$, i.e., $\arg\min_{\mathbf{X} \in \mathcal{X}} \mathrm{cost}(\mathbf{X})$. Although there can be $\exp_q(|\mathbf{g}|)$ possible trial combination, we observe that performance asymptotically converges, so we limit the sampling to finite $n$ trials. A detailed description of our algorithm is provided in Alg. 1.

## 5 EXPERIMENTS

We provide extensive experimental results to demonstrate the superiority of DISCO. We begin by detailing the experimental settings in Sec. 5.1, followed by comparisons with state-of-the-art CO solvers on well-studied TSP problems in Sec. 5.2. Subsequently, we conduct ablations on DISCO components in Sec. 5.3, verify its generalization ability to unseen problem scales in Sec. 5.4, and assess its scalability in solving MIS problems in Sec. 5.5.

### 5.1 EXPERIMENTAL SETTINGS

**Metrics** While DISCO is generically applicable to various NPC problems, our evaluations primarily focus on the most representative TSP problem, as it is a common challenge in the machine learning community with established competitors, providing a solid benchmark to demonstrate our method's superiority. Our evaluation metrics include the average length (Length) of tours and the clock time (Time) required for solving all test instances, presented in seconds (s), minutes (m), or hours (h). We also report the performance gap (Gap), which is the average of the relative decrease in performance compared to a baseline method.

**Baselines** We conduct an extensive comparison of DISCO with a diverse set of baselines, including exact solvers, heuristic solvers, and state-of-the-art learning methods. For exact solvers, our comparisons include Concorde (Applegate et al., 2006) and Gurobi (LLC Gurobi Optimization, 2018). Regarding heuristic solvers, we evaluate against LKH-3 (Helsgaun, 2017), 2-opt (Croes, 1958) and a simple Farthest Insertion principle (Cook et al., 2011). In terms of learning-based methods, we compare with recent advances including AM (Kool et al., 2019), ELG-POMO (Gao et al., 2023), BQ-NCO (Drakulic et al., 2024), and GLOP (Ye et al., 2024b), and diffusion-based solvers DIFUSCO (Sun & Yang, 2023) and T2T (Li et al., 2024). Note that, T2T is currently the most powerful neural solver for TSP problems.

We label the large-scale training instances using the LKH-3 heuristic solver (Helsgaun, 2017) and generate the test instances following the same principle as Fu et al. (2021) and Sun & Yang (2023). All experiments are conducted on a single NVIDIA A100 GPU, paired by AMD EPYC 7662 CPUs @ 2.00GHz. Some learning-based solvers struggle with large problem scales; for instance, Image Diffusion (Graikos et al., 2022) only operates on a $64 \times 64$ greyscale image. To ensure fairness, we compare them on small-scale instances. The results are provided in App. C, along with comparisons with GCN (Joshi et al., 2019), Transformer (Bresson & Laurent, 2021), POMO (Kwon et al., 2020), Sym-NCO (Kim et al., 2022), DPDP (Ma et al., 2021), and MDAM (Xin et al., 2021). Our codes, the mentioned baselines, pre-trained models, and documentation are provided in the Supplementary Material and will be publicly released upon acceptance.

Table 1: Comparisons on large-scale TSP problems. G, S, and BS denotes Greedy decoding, Sampling decoding, and Beam Search (Sutskever et al., 2014), respectively. The symbol * indicates the baseline for computing the performance gap. The symbol † denotes that the diffusion model samples once. N/A indicates that results could not be produced within 24 hours (Qiu et al., 2022), and OOM signifies running out of 80GB GPU memory.

| ALGORITHM | TYPE | TSP-5000 | | | TSP-8000 | | | TSP-10000 | | |
|---|---|---|---|---|---|---|---|---|---|---|
| | | LENGTH ↓ | GAP ↓ | TIME ↓ | LENGTH ↓ | GAP ↓ | TIME ↓ | LENGTH ↓ | GAP ↓ | TIME ↓ |
| CONCORDE | EXACT | N/A | N/A | N/A | N/A | N/A | N/A | N/A | N/A | N/A |
| GUROBI | EXACT | N/A | N/A | N/A | N/A | N/A | N/A | N/A | N/A | N/A |
| LKH-3 (DEFAULT) | HEURISTICS | 51.94* | — | 6.57m | 65.21* | — | 16.23m | 71.77* | — | 8.8h |
| LKH-3 (LESS TRIALS) | HEURISTICS | 52.22 | 0.54% | 5.17m | 66.11 | 1.38% | 13.83m | 71.79 | 0.03% | 51.27m |
| RAW 2-OPT | HEURISTICS | 58.99 | 13.57% | 6.16m | 79.29 | 21.59% | 14.15m | 91.16 | 27.02% | 28.49m |
| FARTHEST INSERTION | HEURISTICS | 57.20 | 10.13% | 0.97m | 72.28 | 10.84% | 5.78m | 80.59 | 12.29% | 13.25m |
| BQ-NCO | RL+G | 175.34 | 237.58% | 75.72m | 725.67 | 1012.82% | 4.98h | | OOM | |
| AM | RL+G | 89.35 | 72.03% | 1.68m | 122.42 | 87.73% | 3.95m | 141.51 | 97.17% | 7.68m |
| ELG-POMO | RL+G | 59.96 | 15.44% | 51.18m | 76.71 | 17.64% | 2.02h | | OOM | |
| GLOP | RL+G | 53.39 | 2.79% | 0.51m | 67.51 | 3.53% | 0.53m | 75.29 | 4.90% | 1.90m |
| DIFUSCO | SL+G† | 53.31 | 2.64% | 8.65m | 67.51 | 3.53% | 19.38m | 73.99 | 3.10% | 35.38m |
| T2T | SL+G† | 53.17 | 2.37% | 25.88m | 67.43 | 3.40% | 1.11h | 73.87 | 2.92% | 1.52h |
| DISCO (**OURS**) | SL+G† | **52.48** | **1.04%** | 5.72m | **66.11** | **1.38%** | 14.32m | **73.85** | **2.90%** | 25.12m |
| AM | RL+BS | 83.93 | 61.59% | 19.07m | 114.82 | 76.08% | 1.13h | 129.40 | 80.28% | 1.81h |
| GLOP | RL+S | 53.28 | 2.58% | 0.54m | 67.41 | 3.37% | 0.59m | 75.27 | 4.88% | 5.96m |
| DIFUSCO | SL+S | 53.15 | 2.33% | 21.07m | 67.41 | 3.37% | 50.18m | 73.90 | 2.97% | 1.83h |
| T2T | SL+S | 53.10 | 2.23% | 47.85m | 67.40 | 3.36% | 1.86h | **73.81** | **2.84%** | 2.47h |
| DISCO (**OURS**) | SL+S | **52.44** | **0.96%** | 9.06m | **66.06** | **1.30%** | 22.82m | **73.81** | **2.84%** | 48.77m |

## 5.2 COMPREHENSIVE COMPARISONS

We compare DISCO to alternative NPC solvers across various large-scale problem instances, including TSP-5000, TSP-8000, and TSP-10000. Given that generating heatmaps with parameterized solvers and transforming them into solutions through decoding strategies has become standard practice (Deudon et al., 2018), we report parameterized solvers' performance decoding with different strategies. Xia et al. (2024) highlight that the MCTS strategy (Fu et al., 2021) heavily relies on TSP-specific heuristics, and is less suited to other problem types. Therefore, we focus on general decoding strategies, including Greedy (Graikos et al., 2022), Sampling (Kool et al., 2019), and 2-opt (Croes, 1958), which represents local search, to evaluate each method's general CO-solving capability. These strategies are introduced in App. F. The performance comparisons with the TSP-specific MCTS strategy can be found in App. H. We align DISCO's decoding settings with DIFUSCO and T2T to demonstrate its superiority as a diffusion solver. To ensure fairness, we apply 2-opt to all learning-based methods, as some solvers like DIFUSCO and T2T use it while others do not. Follow Graikos et al. (2022), we use the Greedy+2-opt strategy by default, and Sampling is conducted 4 times across all problem scales. Unless otherwise noted, DISCO's denoising steps are set to 1 to highlight its efficiency, while DIFUSCO uses 50 steps and T2T uses 20 steps in inference and 3 iterations × 10 steps in gradient search. Additional details are provided in App. G.

The comprehensive results are summarized in Tab. 1. We observe that DISCO outperforms all the previous methods on all problem scales, including T2T which is the current state-of-the-art solver for TSP problems. Diffusion-based methods generally outperform other learning-based approaches, highlighting the significance of diffusion as a choice. Its inherent multi-modal expressiveness makes it particularly well-suited for optimization problems. Notably, beyond its performance advantage, DISCO also demonstrates a significant advantage in inference speed compared to the other two diffusion alternatives, DIFUSCO and T2T, with its inference duration achieving up to 5.28 times speedup, better satisfying many real-world applications that require time-sensitive responses. Since T2T requires gradient-based search during deployment, its computational resource demands are obviously higher than DISCO and DIFUSCO. A detailed comparison is provided in App. H. We also evaluate DISCO on real-world TSP scenarios from TSPLIB (Reinelt, 1991) in App. E. DISCO is the best performer in 28 out of 29 test cases while its inference speed surpasses all compared algorithms, further validating the practicality of our method.

We provide comparisons of DISCO with more recent learning-based methods which are only trainable on small-scale instances in App. C and App. H, with DISCO consistently maintaining its performance advantage. We provide more evidence in App. H to demonstrate the impact of DISCO's multi-modal property on improving solution quality. To facilitate a better understanding of our approach, we provide visual comparisons of denoising results in App. D. These include the evolution of generated heatmaps throughout the denoising process and the correlation between the final solution quality and the total number of diffusion steps. We further test the generalization ability of DISCO as a probabilistic solver to unseen degraded solutions and unseen problem distributions in App. H.

### 5.3 Ablations on DISCO Components

We conduct ablation experiments on two key modules of DISCO: the analytical denoising process and residue constraints. The results are summarized in Tab. 2. We can observe that for the version without these two modules, which can also be regarded as an equal implementation of DIFUSCO, its reverse process requires 50 steps to achieve satisfactory results; otherwise, the solution quality suffers. In contrast, with the analytical denoising process, we can obtain a satisfactory solution with just 1 step, significantly improving inference speed. Moreover, the presence of residue constraints notably enhances the quality of generated heatmaps, as evident from a direct example in Fig. 3. The improvement in predicted heatmap quality naturally translates into higher solution quality, ultimately reflecting the efficacy of DISCO motivation.

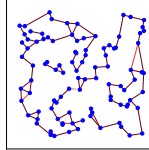 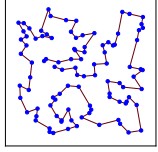

(a) W/o residue.  (b) W/ residue.

Figure 3: Akin to inner loops in a generated TSP heatmap (a), the denoised samples without residues span an enormous NPC space, leading to frequent failures in satisfying problem constraints as (b).

Table 2: Ablations on DISCO components. Step denotes the denoising step number. A and R represent the analytical diffusion process and residue constraints. Note that, DISCO w/o A&R can be regarded as an equal implementation of DIFUSCO.

| | Algorithm | Steps | TSP-8000 | | | TSP-10000 | | |
|---|---|---|---|---|---|---|---|---|
| | | | Length↓ | Gap(%)↓ | Time↓ | Length↓ | Gap(%)↓ | Time↓ |
| Greedy | W/o A&R | 50 | 67.51 | 3.53% | 19.38m | 73.99 | 3.10% | 35.38m |
| | W/o A&R | 1 | 69.43 | 6.47% | 17.23m | 77.67 | 8.22% | 26.80m |
| | W/o R | 1 | 66.65 | 2.21% | 15.85m | 76.27 | 6.27% | 27.27m |
| | DISCO | 1 | **66.11** | **1.38%** | 14.32m | **73.85** | **2.90%** | 25.12m |
| Sampling | W/o A&R | 50 | 67.41 | 3.37% | 50.18m | 73.90 | 2.97% | 1.83h |
| | W/o A&R | 1 | 69.20 | 6.12% | 38.80m | 77.47 | 7.94% | 1.01h |
| | W/o R | 1 | 66.49 | 1.96% | 29.20m | 76.17 | 6.13% | 1.00h |
| | DISCO | 1 | **66.06** | **1.30%** | 22.82m | **73.81** | **2.84%** | 48.77m |

### 5.4 Multi-Modal Graph Search for Generalization

Benefiting from DISCO's verified advantage in both solution quality and inference speed, we can generalize a pre-trained DISCO model $p_{\boldsymbol{\theta}}$ to solve the unseen-scale problem instances off the shelf by a traditional divide-and-conquer strategy. We train the base model $p_{\boldsymbol{\theta}}$ on TSP-100 instances and transfer it to TSP-5000/8000/10000 instances. The decomposed sub-problems $\mathbf{g}$ should jointly cover the global graph problem $G$ at least $\omega = 1$ time. For each sub-problem $g \in \mathbf{g}$, we generate a set of heatmaps $\mathcal{H}$ with $q = 2$ different noises. The results are summarized in Tab. 3. We organize the experiments in the following logic: First, we test diffusion models trained on different problem scales to verify the existence of performance degradation. In addition, we validate that the multi-modal graph search method allows trained models to transfer to unseen problem scales off the shelf. Finally, we propose potential methods to further enhance the performance of our graph search approach.

Table 3: Results on multi-modal graph search. GS means graph search. T indicates training on the corresponding problem scale. 'Best Inter.' refers to selecting the best intermediate $\mathbf{h}$ with the greedily decoded solution from heatmap set $\mathcal{H}$ for each subgraph $g$, rather than random selection to maintain trial diversity.

| Algorithm | Type | Trial | TSP-5000 | | | TSP-8000 | | | TSP-10000 | | |
|---|---|---|---|---|---|---|---|---|---|---|---|
| | | | Length↓ | Gap↓ | Time↓ | Length↓ | Gap↓ | Time↓ | Length↓ | Gap↓ | Time↓ |
| LKH-3 (default) | Heuristics | 10000 | 51.94* | — | 6.57m | 65.21* | — | 16.23m | 71.77* | — | 8.8h |
| Att-GCN | SL+MCTS | 1 | 52.76 | 1.58% | 13.30m | 66.77 | 2.40% | 25.95m | 74.60 | 4.86% | 37.97m |
| GLOP | RL | 1 | 53.39 | 2.79% | 0.51m | 67.51 | 3.53% | 0.53m | 75.29 | 4.90% | 1.90m |
| DISCO (TSP-5000, T) | SL+G† | 1 | **52.48** | **1.04%** | 5.72m | 67.42 | 3.39% | 17.52m | 74.98 | 4.47% | 25.37m |
| DISCO (TSP-8000, T) | SL+G† | 1 | 52.97 | 1.98% | 5.10m | **66.11** | **1.38%** | 17.32m | 74.60 | 3.94% | 25.70m |
| DISCO (TSP-10000, T) | SL+G† | 1 | 53.21 | 2.44% | 5.82m | 67.27 | 3.16% | 17.46m | **73.85** | **2.90%** | 25.12m |
| DIFUSCO (Best Inter.) | SL+GS+G† | 1 | 52.78 | 1.62% | 1.31h | 66.86 | 2.53% | 2.16h | 74.33 | 3.57% | 4.91h |
| DIFUSCO | SL+GS+G† | 50 | 52.67 | 1.41% | 2.11h | 66.61 | 2.15% | 4.81h | 74.35 | 3.60% | 5.93h |
| DISCO (Best Inter.) | SL+GS+G† | 1 | 52.77 | 1.60% | 8.12m | 66.56 | 2.07% | 19.82m | 74.45 | 3.73% | 36.43m |
| DISCO | SL+GS+G† | 50 | **52.65** | **1.37%** | 32.40m | **66.52** | **2.01%** | 1.34h | **74.24** | **3.44%** | 1.82h |
| DISCO | SL+GS+G† | 100 | 52.62 | 1.31% | 57.53m | 66.52 | 2.01% | 2.31h | 74.22 | 3.41% | 3.76h |
| DISCO ($\omega = 4$) | SL+GS+G† | 50 | 52.60 | 1.27% | 35.45m | 66.48 | 1.95% | 1.40h | 74.23 | 3.43% | 2.18h |
| DISCO | SL+GS+MCTS | 50 | **52.32** | **0.73%** | 41.98m | **66.12** | **1.40%** | 1.59h | **73.69** | **2.68%** | 2.10h |

We can observe that when testing a trained model on a different problem scale, although DISCO generalizes decently, it performs less effectively compared to models trained on the equivalent scale. Meanwhile, our multi-modal graph search algorithm, combined with $p_{\boldsymbol{\theta}}$ trained only on TSP-100, exhibits better performance than direct generalization. Att-GCN (Fu et al., 2021) and GLOP (Ye et al., 2024b) also adopt the divide-and-conquer mechanism for generalization. However, their parameterized solver lacks the ability to generate diverse trials, which causes the final solution may get stuck in the sub-optimum. Our DISCO method increases the diversity of solution samples and broadens exploration in the solution space, enhancing the likelihood of finding higher-quality

solutions and being more effective than Att-GCN and GLOP. Although DIFUSCO also possesses the multi-modal property, its inference speed is prohibitively slow. DISCO achieves at least 3.26 times faster inference speed than DIFUSCO for graph search, while also delivering superior results.

While our graph search approach offers $\exp_q(|\mathbf{g}|)$ possibilities for enriching solution diversity, its performance asymptotically converges to the number of sampled trials, as confirmed by the generalization results on TSP-1000 in Fig. 4. The required trial number increases with solution variance, which is controlled by the number of denoising steps. We recommend 2-step denoising for better practice. We also compare our method with a variant that does not generate diverse trials—specifically, selecting the best intermediate $\mathbf{h}$ with the greedily decoded solution from heatmap set $\mathcal{H}$ for each subgraph $g$—and find that this version generally performs worse. This amplifies the importance of solution diversity for solving CO problems. DISCO's performance can be further enhanced by trading off time costs through various means such as increasing sampled trials, augmenting the subgraph number $|\mathbf{g}|$ by controlling $\omega$, and re-decoding the merged heatmap combinations corresponding to the most promising trial with more sophisticated strategies like MCTS. These enhancements can even lead to better performance than models trained on the corresponding scale. These conclusions are corroborated in Table 3.

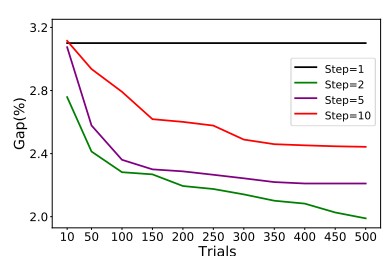

Figure 4: Asymptotic performance of multi-modal graph search with trial number.

Table 4: Results on MIS problems. TS denotes tree search.

| METHOD | TYPE | SATLIB | | | ER-[700-800] | | |
|---|---|---|---|---|---|---|---|
| | | SIZE ↑ | GAP ↓ | TIME ↓ | SIZE ↑ | GAP ↓ | TIME ↓ |
| KaMIS | HEURISTICS | 425.96* | — | 37.58m | 44.87* | — | 52.13m |
| GUROBI | EXACT | 425.95 | 0.00% | 26.00m | 41.38 | 7.78% | 50.00m |
| DGL | SL+TS | N/A | N/A | N/A | 37.26 | 16.96% | 22.71m |
| INTEL | SL+TS | N/A | N/A | N/A | 38.80 | 13.43% | 20.00m |
| INTEL | SL+G | 420.66 | 1.48% | 23.05m | 34.86 | 22.31% | 6.06m |
| DIMES | RL+G | 421.24 | 1.11% | 24.17m | 38.24 | 14.78% | 6.12m |
| DIFUSCO | SL+G | 424.50 | 0.34% | 13.00m | 38.83 | 12.40% | 8.80m |
| T2T | RL+G | **425.02** | **0.22%** | 14.30m | 39.56 | 11.83% | 8.53m |
| DISCO (**OURS**) | SL+G | 424.58 | 0.32% | 10.32m | **40.30** | **10.17%** | 9.00m |
| LwD | RL+S | 422.22 | 0.88% | 18.83m | 41.17 | 8.25% | 6.33m |
| DIMES | RL+S | 423.28 | 0.63% | 20.26m | 42.06 | 6.26% | 12.01m |
| GFLOWNET | UL+S | 423.54 | 0.57% | 23.22m | 41.14 | 8.53% | 2.92m |
| DIFUSCO | SL+S | 425.04 | 0.22% | 26.09m | 40.70 | 9.29% | 17.33m |
| T2T | SL+S | **425.06** | **0.21%** | 24.56m | 41.37 | 7.81% | 29.73m |
| DISCO (**OURS**) | SL+S | **425.06** | **0.21%** | 25.38m | **42.21** | **5.93%** | 16.93m |

## 5.5 EVALUATIONS ON MAXIMAL INDEPENDENT SET

Besides TSP, we evaluate DISCO on commonly studied MIS problems, both of which are adequately representative of edge-based and node-based NPC problems. Evaluations are conducted on SATLIB (Hoos & Stützle, 2000) and Erdős-Rényi (ER) (Erdős & Rényi, 1960) graph sets, which exhibit challenge for recent learning-based solvers (Li et al., 2018; Ahn et al., 2020; Böther et al., 2022; Qiu et al., 2022; Zhang et al., 2023). Training instances are labeled using the KaMIS heuristic solver (Lamm et al., 2016), with test instances aligned with Qiu et al. (2022). We adopt the same 50 denoising steps and 4 sample times as DIFUSCO to distinguish model capabilities. Details of experimental settings and baselines can be found in App. B. We report the average size of the independent set (Size) in Tab. 4. DISCO exhibits a clear performance advantage over most competitors.

## 6 CONCLUSION

We propose DISCO, an efficient diffusion solver for large-scale CO problems. DISCO obtains improved solution quality by restricting the sampling space to a more meaningful domain guided by solution residues, and enables rapid solution generation with minimal denoising steps. DISCO delivers strong performance on large-scale TSP instances and challenging MIS benchmarks SATLIB and Erdős-Rényi, with inference duration up to $5.28$ times faster than existing diffusion solver alternatives. Through further combining a traditional divide-and-conquer strategy, DISCO can be generalized to solve unseen-scale problem instances off the shelf, even outperforming models trained specifically on those scales.

This work has two limitations. First, DISCO relies on supervised learning and decoding strategies to transform output heatmaps, limiting it to offline operations where iterative optimization is feasible. For online operations, DISCO must generate immediate high-quality solutions across the NPC problem space. Future work should explore integrating trial-and-error methods for online applications. Second, DISCO's multi-modal graph search can lead to exponential growth in trial variance. While this variance aids in exploring the solution space and finding optimal solutions, it also increases computational costs. Developing a lightweight policy for smarter trail sampling is promising.

## 7 CODE OF ETHICS

Our proposed DISCO method is a general-purpose parameterized solver for CO problems. DISCO leverages diffusion technologies to address the multi-modal nature of CO problems effectively. DISCO optimizes both its forward and reverse processes more efficiently for solution generation, and significantly excels in both inference speed and solution quality. This improved efficiency further enhances DISCO's capabilities to generalize to arbitrary-scale instances off the shelf. We believe that such efficient, learnable neural solvers for NPC problems will have a positive impact on a broad range of real-world applications (Ghiani et al., 2003; Xu et al., 2018).

## 8 REPRODUCIBILITY

We provide detailed descriptions of the experiment settings in Sec. 5.1, more implementation details can be found in App. B and App. G. The code for DISCO, the mentioned baselines, pre-trained models, and detailed accompanying documentation are also included in the Supplementary Material to ensure reproducibility.

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

APPENDIX

## A   EQUIVALENCE ANALYSIS BETWEEN DDM AND DISCO

Real-world combinatorial optimization (CO) problems often require the rapid generation of high-quality solutions $\mathbf{X}_s$ for problem instance $s$. Previous neural solvers have suffered from the expressiveness limitation when confronted with multiple optimal solutions for the same graph (Khalil et al., 2017; Gu et al., 2018; Li et al., 2018). In contrast, diffusion probabilistic models (DPMs) (Ho et al., 2020) have shown promising prospects for generating a wide variety of distributions suitable for CO solving. An obvious bottleneck is the slow inference speed of DPMs. This is due to DPMs employing numerical integration during the reverse process, requiring multiple steps of accumulation and solving and significantly incurring time overhead.

Inspired by decoupled diffusion models (DDMs) (Huang et al., 2023a), we substitute the time-consuming numerical integration process with an analytically solvable form. The original solution-to-noise mapping can be decoupled into solution-to-zero and zero-to-noise mapping:

$$\mathbf{x}_t = \mathbf{x}_0 + \int_0^t \mathbf{f}_t dt + \int_0^t d\mathbf{w}_t, \quad \mathbf{x}_0 \sim q(\mathbf{x}_0), \tag{10}$$

where $\mathbf{x}_0 + \int_0^t \mathbf{f}_t dt$ describes the solution attenuation and $\int_0^t d\mathbf{w}_t$ describes the noise accumulation. Since $\mathbf{f}_t$ can be designed analytically, the efficiency of the reversed process can be improved by much fewer evaluation steps, e.g., inference steps = 1 or 2. The distribution of $\mathbf{x}_t$ conditioned on $\mathbf{x}_0$ is defined as:

$$q(\mathbf{x}_t \mid \mathbf{x}_0) = \mathcal{N}(\mathbf{x}_t; \mathbf{x}_0 + \mathbf{F}_t, t\mathbf{I}), \tag{11}$$

where $\mathbf{F}_t = \int_0^t \mathbf{f}_t d_t$ and we sample $\mathbf{x}_t$ by $\mathbf{x}_t = \mathbf{x}_0 + \mathbf{F}_t + \sqrt{t}\boldsymbol{\epsilon}$ with $\boldsymbol{\epsilon} \sim \mathcal{N}(\mathbf{0}, \mathbf{I})$.

For a reverse time, the sampling formula for $\mathbf{x}_0$ is based on the analytic attenuation function $\mathbf{f}_t$ that models image to zero transition. We employ continuous-time Markov chain with the smallest time step $\delta t \to 0^+$ and use conditional distribution $q(\mathbf{x}_{t-\Delta t} \mid \mathbf{x}_t, \mathbf{x}_0)$ to approximate $q(\mathbf{x}_{t-\Delta t} \mid \mathbf{x}_t, \mathbf{x}_0)$.

$$q(\mathbf{x}_{t-\Delta t} \mid \mathbf{x}_t, \mathbf{x}_0) \propto \exp\left\{ -\frac{(\mathbf{x}_{t-\Delta t} - \widetilde{\mathbf{u}})^2}{2\widetilde{\sigma}^2 \mathbf{I}} \right\}, \tag{12}$$

where $\boldsymbol{\epsilon} \sim \mathcal{N}(\mathbf{0}, \mathbf{I})$, $\widetilde{\mathbf{u}} = \mathbf{x}_t + \mathbf{F}_{t-\Delta t} - \mathbf{F}_t + \boldsymbol{\epsilon}\Delta t / \sqrt{t}$, and $\widetilde{\sigma}^2 = \Delta t(t - \Delta t)/t$. Since $\mathbf{f}_t$ has an analytic form, we can avoid the ordinary differential equation-based denoising and instead directly sample $\mathbf{x}_0$ with an arbitrary step size, which significantly reduces the inference time.

Although the inference speed can benefit from the analytically solvable form, denoising methods still require inefficient sampling from the entire NPC solution space of CO problems, which typically grows exponentially with the number of problem scale $N$. We propose to constrain the sampling space into a more meaningful one by introducing residues (Liu et al., 2024) to DDM, which is our **DISCO** method, i.e., an efficient **DI**ffusion **S**olver for large-scale **CO** problems. The reversed process starts from both noise and an exceedingly cost-effective degraded solution, confining the generation process in a more meaningful and smaller domain close to the high-quality labels. The residue prioritizes certainty while the noise emphasizes diversity, so that to ensure solution effectiveness while still maintaining their multi-modal property of output distributions.

Instead of the traditional forward process merely outputting noise, it is now a combination of noise $\mathbf{x}_t$ and a degraded solution $\mathbf{X}_d$ for generating residue constraints $\mathbf{x}_{res} = \mathbf{X}_d - \mathbf{x}_0$. The degraded solution $\mathbf{X}_d$ is an exceedingly cost-effective path but satisfies problem constants. For example, $0 - 1 - \ldots - n - 0$, connecting nodes in sequential order. Since DDM has already demonstrated its equivalence to previous diffusion processes defined by Equation 3 (Huang et al., 2023a), we now provide proof of the equivalence between our method and DDM to demonstrate the effectiveness of DISCO in a theoretical aspect.

**Forward Process**   The proposed forward formula considering residue is:

$$\mathbf{x}_t = \mathbf{x}_0 + \int_0^t \mathbf{x}_{res} dt + \sqrt{t}\boldsymbol{\epsilon}. \tag{13}$$

Compared with the original forward formulation of DDM (Eq. 10), the proposed forward formulation utilizes a different function $\mathbf{x}_{res}$ substituting the attenuation function $\mathbf{f}_t$, which means the two diffusion processes are equivalent.

**Reversed Process**  In the reversed process, we need to parameterize two components: $\mathbf{x}_{res}^{\boldsymbol{\theta}}$ and $\boldsymbol{\epsilon}^{\boldsymbol{\theta}}$, which estimate the residue $\mathbf{x}_{res}$ and the noise $\boldsymbol{\epsilon}$, respectively. From Eq. 6, we have:

$$\mathbf{x}_0 = \mathbf{x}_t - \int_0^t \mathbf{x}_{res}dt - \sqrt{t}\boldsymbol{\epsilon}. \tag{14}$$

Thus, the reverse process can be defined as:

$$p_\theta(\mathbf{x}_{t-\Delta t} \mid \mathbf{x}_t) := q(\mathbf{x}_{t-\Delta t} \mid \mathbf{x}_t, \mathbf{x}_0). \tag{15}$$

Applying Bayes' theorem (Jaynes, 2003), we obtain:

$$q(\mathbf{x}_{t-\Delta t} \mid \mathbf{x}_t, \mathbf{x}_0) = \frac{q(\mathbf{x}_t \mid \mathbf{x}_{t-\Delta t})q(\mathbf{x}_{t-\Delta t} \mid \mathbf{x}_t)}{q(\mathbf{x}_t \mid \mathbf{x}_0)} \tag{16}$$

$$= \frac{q(\mathbf{x}_t \mid \mathbf{x}_{t-\Delta t})\mathcal{N}(\mathbf{x}_{t-\Delta t}; \mathbf{x}_0 + \mathbf{H}_{t-\Delta t}, (t-\Delta t)\mathbf{I})}{\mathcal{N}(\mathbf{x}_t; \mathbf{x}_0 + \mathbf{F}_t, t\mathbf{I})}.$$

Eq. 16 aligns with the reverse process in DDM, thus the reverse process formula is:

$$q(\mathbf{x}_{t-\Delta t} \mid \mathbf{x}_t, \mathbf{x}_0) \propto \exp\left(-\frac{(\mathbf{x}_{t-\Delta t} - \mathbf{u})^2}{2\sigma^2\mathbf{I}}\right), \tag{17}$$

where $\widetilde{\mathbf{u}} = \mathbf{x}_t - \int_{t-\Delta t}^t h_t dt - \frac{\Delta t}{\sqrt{t}}\boldsymbol{\epsilon}, \widetilde{\sigma}^2 = \frac{\Delta t(t-\Delta t)}{t}$, equivalent to the reverse formula of DDM.

# B  MAXIMAL INDEPENDENT SET

Besides the TSP problem, we also evaluate DISCO on commonly studied MIS problems, both of which are adequately representative of edge-based and node-based NPC problems. We give specific details of experimental settings in this section.

**Datasets**  We conduct evaluations on SATLIB (Hoos & Stützle, 2000) and Erdős-Rényi (ER) (Erdős & Rényi, 1960) graph sets, which exhibit challenge for recent learning-based solvers (Li et al., 2018; Ahn et al., 2020; Böther et al., 2022; Qiu et al., 2022; Zhang et al., 2023). The training instances are labeled by the KaMIS heuristic solver (Lamm et al., 2016). The split of test instances on SAT datasets and the random-generated ER test graphs are taken from (Qiu et al., 2022).

**Metrics**  We compare the performance of different probabilistic solvers by the average size (Size) of the predicted maximal independent set for each test instance; larger sizes indicate better performance. We also use the same Gap and Time definitions as in the TSP case. We adopt the same denoising steps and sample times as DIFUSCO (Sun & Yang, 2023) to distinguish model capabilities. Specifically, we use 50 steps for denoising heatmap and generate 4 times for sampling strategies. Following the principle of efficiency, we randomly sample a set of nodes from the original graph with a probability of 50% to obtain the degraded solution.

**Baselines**  We compare DISCO with 9 other MIS solvers on the same test sets, including two traditional OR methods and seven learning-based approaches. For the traditional methods, we use Gurobi and KaMIS (Lamm et al., 2016) as baselines. For the learning-based methods, we compare with LwD (Ahn et al., 2020), Intel (Li et al., 2018), DGL (Böther et al., 2022), DIMES (Qiu et al., 2022), GFlowNet (Zhang et al., 2023), DIFUSCO (Sun & Yang, 2023), and T2T (Li et al., 2024).

# C  COMPARISONS ON SMALL-SCALE TSP INSTANCES

Some learning-based solvers struggle with large problem scales, we compare them on small-scale instances for fairness. Specifically, we compare with learning-based methods Image Diffusion (Graikos et al., 2022), GCN (Joshi et al., 2019), Transformer (Bresson & Laurent, 2021), POMO (Kwon et al., 2020), Sym-NCO (Kim et al., 2022), DPDP (Ma et al., 2021), and MDAM (Xin et al., 2021). Along with the learning-based methods works on large scales including AM (Kool et al., 2019), DIFUSCO (Sun & Yang, 2023), and T2T (Li et al., 2024). We compare with traditional operation methods including Concorde (Applegate et al., 2006) and Gurobi (LLC Gurobi Optimization, 2018),

Table 5: Comparisons on TSP-50 and TSP-100. The symbol ∗ denotes the baseline for computing the performance gap. The symbol † indicates that the diffusion model samples once.

| ALGORITHM | TYPE | TSP-50 LENGTH↓ | TSP-50 GAP(%)↓ | TSP-100 LENGTH↓ | TSP-100 GAP(%)↓ |
|---|---|---|---|---|---|
| CONCORDE | EXACT | 5.69* | 0.00 | 7.76* | 0.00 |
| 2-OPT | HEURISTICS | 5.86 | 2.95 | 8.03 | 3.54 |
| AM | GREEDY | 5.80 | 1.76 | 8.12 | 4.53 |
| GCN | GREEDY | 5.87 | 3.10 | 8.41 | 8.38 |
| TRANSFORMER | GREEDY | 5.71 | 0.31 | 7.88 | 1.42 |
| POMO | GREEDY | 5.73 | 0.64 | 7.84 | 1.07 |
| SYM-NCO | GREEDY | - | - | 7.84 | 0.94 |
| DPDP | $1k$-IMPROVEMENTS | 5.70 | 0.14 | 7.89 | 1.62 |
| IMAGE DIFFUSION | GREEDY† | 5.76 | 1.23 | 7.92 | 2.11 |
| DIFUSCO | GREEDY† | 5.70 | 0.10 | 7.78 | 0.24 |
| T2T | GREEDY† | **5.69** | **0.04** | **7.77** | **0.18** |
| DISCO (**OURS**) | GREEDY† | 5.70 | 0.16 | 7.80 | 0.58 |
| AM | $1k\times$SAMPLING | 5.73 | 0.52 | 7.94 | 2.26 |
| GCN | $2k\times$SAMPLING | 5.70 | 0.01 | 7.87 | 1.39 |
| TRANSFORMER | $2k\times$SAMPLING | 5.69 | 0.00 | 7.76 | 0.39 |
| POMO | $8\times$AUGMENT | 5.69 | 0.03 | 7.77 | 0.14 |
| SYM-NCO | $100\times$SAMPLING | - | - | 7.79 | 0.39 |
| MDAM | $50\times$SAMPLING | 5.70 | 0.03 | 7.79 | 0.38 |
| DPDP | $100k$-IMPROVEMENTS | 5.70 | 0.00 | 7.77 | 0.00 |
| DIFUSCO | $16\times$SAMPLING | **5.69** | **-0.01** | **7.76** | **-0.01** |
| T2T | $16\times$SAMPLING | **5.69** | **-0.01** | **7.76** | **-0.01** |
| DISCO (**OURS**) | $16\times$SAMPLING | **5.69** | **-0.01** | **7.76** | 0.03 |

LKH-3 (Helsgaun, 2017), 2-OPT (Croes, 1958), and Farthest Insertion (Cook et al., 2011). We label the training instances using the Concorde solver for TSP-50/100 and we take the same test instances as (Kool et al., 2019; Joshi et al., 2022). The comprehensive results are summarized in Tab. 5, with DISCO consistently maintaining its performance advantage. We visualize the corresponding denoising processes in Fig. 5.

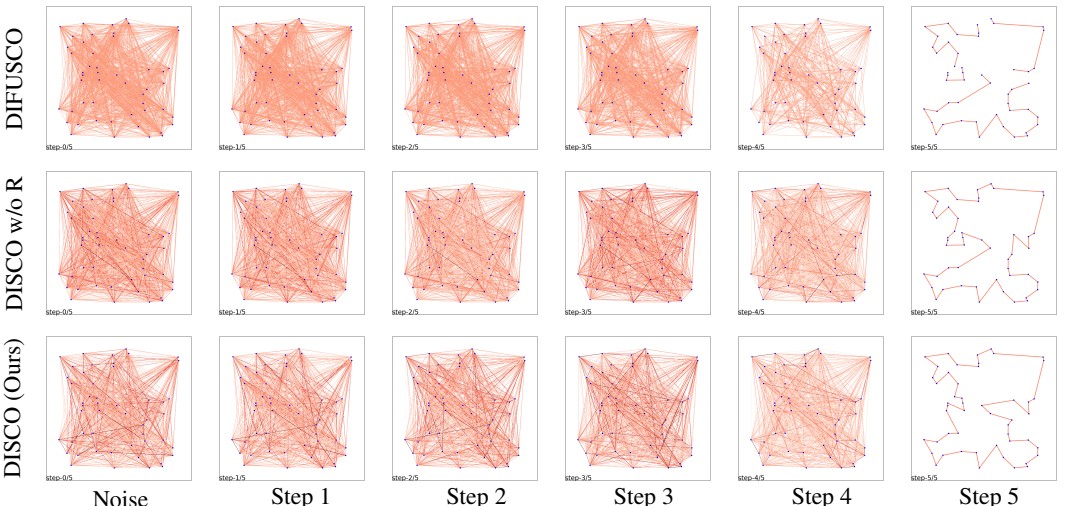

Figure 5: Denoising processes on TSP-50. Decoding the final heatmap with Greedy+2-opt yields tour lengths of **5.95** for DIFUSCO, **5.77** for DISCO without residues (w/o R), and **5.75** for DISCO.

# D QUALITATIVE RESULTS

## D.1 DENOISING PROCESSES ON LARGE-SCALE INSTANCES

We illustrate the denoising processes of different diffusion methods on large-scale problems in Figure 6, using TSP-1000 as an example for clarity. The analytical denoising design and introduction of residues in DISCO ensure that high-quality solutions can be obtained with a few steps. In contrast,

alternative methods generate solutions that frequently violate problem constraints, such as isolated nodes, non-closed tours, and inner loops.

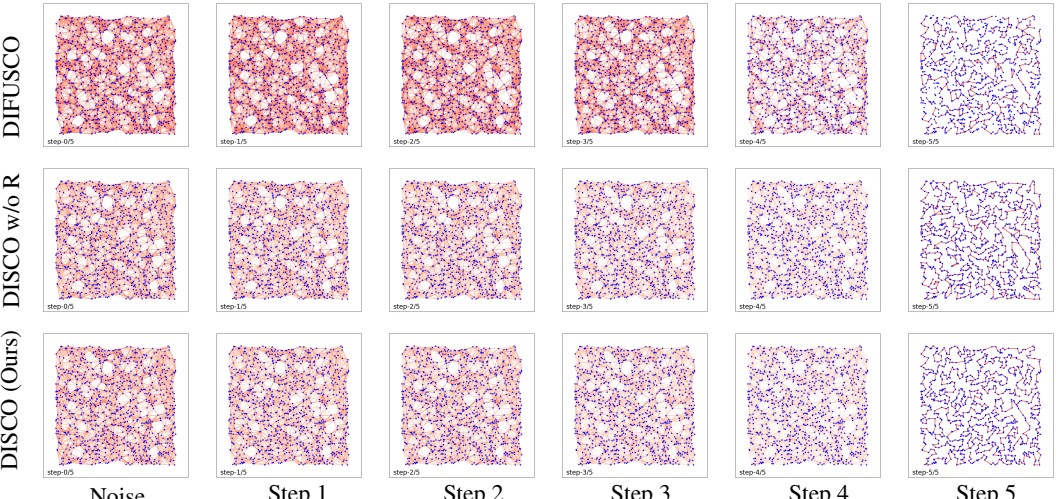

Figure 6: Denoising processes on TSP-1000. Decoding the final heatmap with Greedy+2-opt yields tour lengths of **27.69** for DIFUSCO, **26.33** for DISCO without residues (w/o R), and **25.35** for DISCO.

## D.2 Performance with Different Denoising Steps

Here, we demonstrate the final heatmaps $x_0$ generated by different denoising steps with different diffusion methods. The visualizations are summarized in Fig. 7, which still opt for TSP-1000 to ensure readability. The corresponding decoded tour lengths are annotated directly below each plot. It is evident that DISCO consistently produces high-quality solutions across various denoising steps. Particularly for time-sensitive scenarios requiring few denoising steps, DISCO maintains a significant advantage over the baselines.

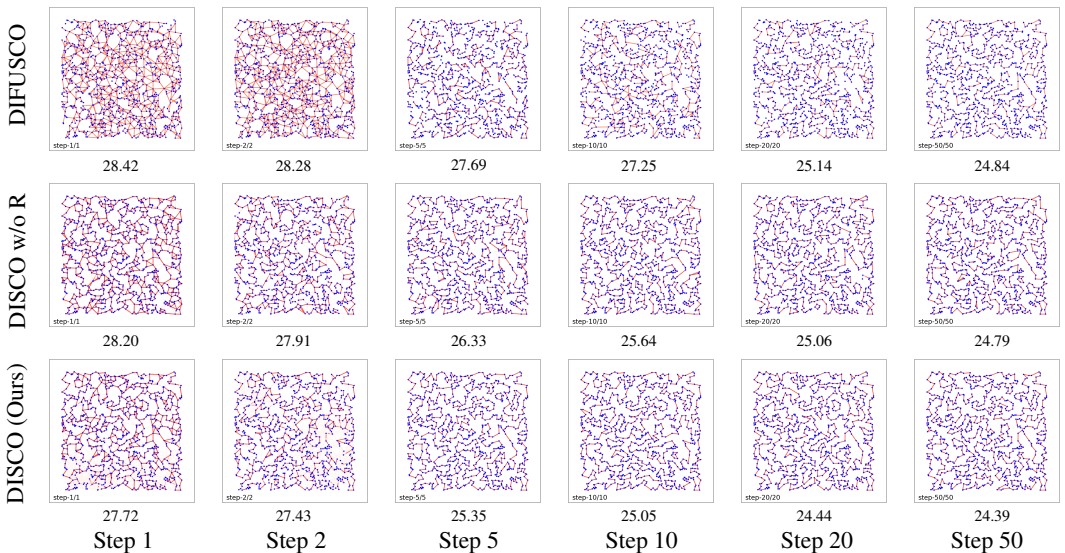

Figure 7: Generated heatmaps on TSP 1000-instances under different denoising steps, with the final decoded tour length captioned below. DISCO consistently produces higher-quality heatmaps that better satisfy the problem constraints, leading to better decoding results with the same number of denoising steps.

# E   GENERALIZATION TO REAL-WORLD INSTANCES

We evaluate DISCO on TSPLIB (Reinelt, 1991), a collection of real-world TSP scenarios, to assess its effectiveness in transferring knowledge from the synthetically generated data to the real world. We directly transfer the DISCO models trained on TSP-50 and TSP-100 to these real-world instances without any fine-tuning. Each instance strictly follows the evaluation protocol proposed by TSPLIB. The results are summarized in Tab. 6. We can observe that DISCO is the best performer in 28 out of 29 test cases. Notably, as a diffusion-based algorithm, DISCO's solving speed is the fastest among all compared algorithms across all cases, further demonstrating its efficiency advantage and practicality. The test code and models for this part are provided in the Supplementary Material for reproducibility. We also provide visualizations of each solution generated by DISCO in Fig. 8.

Table 6: TSPLIB performance. We indicate the training scales for DISCO/DIFUSCO in parentheses.

| Method | Kool et al. | | Joshi et al. | | d O Costa et al. | | Hudson et al. | | DIFUSCO (50) | | DIFUSCO (100) | | DISCO (50) | | DISCO (100) | |
|---|---|---|---|---|---|---|---|---|---|---|---|---|---|---|---|---|
| Instance | Time (s) | Gap (%) | Time (s) | Gap (%) | Time (s) | Gap (%) | Time (s) | Gap (%) | Time (s) | Gap (%) | Time (s) | Gap (%) | Time (s) | Gap (%) | Time (s) | Gap (%) |
| eil51 | 0.125 | 1.628 | 3.026 | 8.339 | 28.051 | 0.067 | 10.074 | **0.000** | 0.482 | **0.000** | 0.519 | 0.117 | **0.049** | **0.000** | **0.049** | **0.000** |
| berlin52 | 0.129 | 4.169 | 3.068 | 33.225 | 31.874 | 0.449 | 10.103 | 0.142 | 0.527 | **0.000** | 0.526 | **0.000** | 0.047 | **0.000** | 0.048 | 0.000 |
| st70 | 0.200 | 1.737 | 4.037 | 24.785 | 23.964 | 0.040 | 10.053 | 0.764 | 0.663 | 0.000 | 0.670 | 0.000 | 0.062 | -0.741 | 0.064 | -0.741 |
| eil76 | 0.225 | 1.992 | 4.303 | 27.411 | 26.551 | 0.096 | 10.155 | 0.163 | 0.788 | 0.000 | 0.788 | 0.174 | 0.076 | -0.557 | 0.072 | -0.557 |
| pr76 | 0.226 | 0.816 | 4.378 | 27.793 | 39.485 | 1.228 | 10.049 | 0.039 | 0.765 | 0.000 | 0.785 | 0.187 | 0.074 | -0.379 | 0.072 | -0.379 |
| rat99 | 0.347 | 2.645 | 5.559 | 17.633 | 32.188 | 0.123 | 9.948 | 0.550 | 1.236 | 1.187 | 1.192 | 0.000 | 0.115 | 0.000 | 0.103 | -0.165 |
| kroA100 | 0.352 | 4.017 | 5.705 | 28.828 | 42.095 | 18.313 | 10.255 | 0.728 | 1.259 | 0.741 | 1.217 | 0.000 | 0.110 | -0.019 | 0.106 | -0.019 |
| kroB100 | 0.352 | 5.142 | 5.712 | 34.686 | 35.137 | 1.119 | 10.317 | 0.147 | 1.252 | 0.648 | 1.235 | 0.742 | 0.116 | 0.235 | 0.108 | 0.262 |
| kroC100 | 0.352 | 0.972 | 5.641 | 35.506 | 34.333 | 0.349 | 10.172 | 1.571 | 1.199 | 1.712 | 1.168 | 0.000 | 0.108 | 0.029 | 0.103 | -0.067 |
| kroD100 | 0.352 | 2.717 | 5.621 | 38.018 | 25.772 | 0.866 | 10.375 | 0.572 | 1.226 | 0.000 | 1.175 | 0.000 | 0.118 | -0.117 | 0.110 | -0.117 |
| kroE100 | 0.352 | 1.470 | 5.650 | 26.589 | 34.475 | 1.832 | 10.270 | 1.216 | 1.208 | 0.274 | 1.197 | 0.274 | 0.114 | 0.168 | 0.110 | 0.000 |
| rd100 | 0.352 | 3.407 | 5.737 | 50.432 | 28.963 | 0.003 | 10.125 | 0.459 | 1.191 | 0.000 | 1.172 | 0.000 | 0.101 | -0.733 | 0.097 | -0.733 |
| eil101 | 0.359 | 2.994 | 5.790 | 26.701 | 23.842 | 0.387 | 10.276 | 0.201 | 1.222 | 0.576 | 1.215 | 0.000 | 0.114 | -0.318 | 0.107 | -0.318 |
| lin105 | 0.380 | 1.739 | 5.938 | 34.902 | 39.517 | 1.867 | 10.330 | 0.606 | 1.321 | 0.000 | 1.280 | 0.000 | 0.124 | -0.306 | 0.107 | -0.306 |
| pr107 | 0.391 | 3.933 | 5.964 | 80.564 | 29.039 | 0.898 | 9.977 | 0.439 | 1.381 | 0.228 | 1.378 | 0.415 | 0.148 | -0.199 | 0.144 | -0.169 |
| pr124 | 0.499 | 3.677 | 7.059 | 70.146 | 29.570 | 10.322 | 10.360 | 0.755 | 1.803 | 0.925 | 1.782 | 0.494 | 0.144 | 0.198 | 0.144 | 0.151 |
| bier127 | 0.522 | 5.908 | 7.242 | 45.561 | 39.029 | 3.044 | 10.260 | 1.948 | 1.938 | 1.011 | 1.915 | 0.366 | 0.176 | -0.379 | 0.169 | -1.026 |
| ch130 | 0.550 | 3.182 | 7.351 | 39.090 | 34.436 | 0.709 | 10.032 | 3.519 | 1.989 | 1.970 | 1.967 | 0.077 | 0.153 | 0.245 | 0.162 | -0.016 |
| pr136 | 0.585 | 5.064 | 7.727 | 58.673 | 31.056 | 0.000 | 10.379 | 3.387 | 2.184 | 2.490 | 2.142 | 0.000 | 0.146 | 0.069 | 0.180 | -0.342 |
| pr144 | 0.638 | 7.641 | 8.132 | 55.837 | 28.913 | 1.526 | 10.276 | 3.581 | 2.478 | 0.519 | 2.446 | 0.261 | **0.159** | -0.063 | 0.186 | -0.063 |
| ch150 | 0.697 | 4.584 | 8.546 | 49.743 | 35.497 | 0.312 | 10.109 | 2.113 | 2.608 | 0.376 | 2.555 | 0.000 | **0.169** | 0.276 | 0.202 | -0.061 |
| kroA150 | 0.695 | 3.784 | 8.450 | 45.411 | 29.399 | 0.724 | 10.331 | 2.984 | 2.617 | 3.753 | 2.601 | 0.000 | 0.174 | 0.033 | 0.208 | -0.098 |
| kroB150 | 0.696 | 2.437 | 8.573 | 56.745 | 29.005 | 0.886 | 10.018 | 3.258 | 2.626 | 1.839 | 2.592 | **0.067** | **0.176** | 0.554 | 0.206 | 0.417 |
| pr152 | 0.708 | 7.494 | 8.632 | 33.925 | 29.003 | 0.029 | 10.267 | 3.119 | 2.716 | 1.751 | 2.712 | 0.481 | **0.183** | 0.122 | 0.221 | -0.062 |
| u159 | 0.764 | 7.551 | 9.012 | 38.338 | 28.961 | 0.054 | 10.428 | 1.020 | 2.963 | 3.758 | 2.892 | 0.000 | 0.184 | -0.067 | 0.196 | -0.067 |
| rat195 | 1.114 | 6.893 | 11.236 | 24.968 | 34.425 | 0.743 | 12.295 | 1.666 | 4.400 | 1.540 | 4.428 | 0.767 | **0.266** | 0.947 | 0.310 | 0.560 |
| d198 | 1.153 | 373.020 | 11.519 | 62.351 | 30.864 | 0.522 | 12.596 | 4.772 | 4.615 | 4.832 | 4.153 | 3.337 | **0.297** | 0.330 | 0.375 | 0.292 |
| kroA200 | 1.150 | 7.106 | 11.702 | 40.885 | 33.832 | 1.441 | 11.088 | 2.029 | 4.710 | 6.187 | 4.686 | 0.065 | **0.301** | 1.134 | 0.346 | -0.398 |
| kroB200 | 1.150 | 8.541 | 11.689 | 43.643 | 31.951 | 2.064 | 11.267 | 2.589 | 4.606 | 6.605 | 4.619 | 0.590 | **0.301** | 1.481 | 0.346 | 0.065 |
| Mean | 0.532 | 16.767 | 7.000 | 40.025 | 31.766 | 1.725 | 10.420 | 1.529 | 1.999 | 1.480 | 1.966 | 0.290 | **0.149** | 0.067 | 0.161 | -0.136 |

# F   DECODING STRATEGY

For offline problems (Papadimitriou & Steiglitz, 1998; Martello et al., 2000), all input data and all constraints are fully provided before solving the problems. The decision-makers can fully utilize all relevant information for comprehensive analysis and iteratively improve solution quality. Providing an initial solution by SL and further refining it by decoding strategies has become a common practice (Deudon et al., 2018). We introduce the following decoding strategies combined with DISCO, including Greedy (Graikos et al., 2022), Sampling (Kool et al., 2019), and 2-OPT strategies (Croes, 1958).

**Greedy Strategy**   We use a straightforward greedy strategy to decode solutions from heatmaps produced by probabilistic models. Specifically, we iteratively add the highest-scoring candidates among the remaining ones to the partial solution. We repeat this process until all relevant nodes/edges are incorporated. For diffusion-based methods, we sample the initial solution once with a single noise data $\mathbf{x}_t$. We set the denoising step as 1 for DISCO when executing this greedy strategy, allowing the variance $\sigma$ to approach zero (as described in Eq. 7) to generate more confident solutions.

**Sampling Strategy**   Probabilistic solvers usually sample multiple solutions (Kool et al., 2019) through various means and execute the best one. Statistically, increasing the number of samples can enhance the breadth and depth while exploring the solution space, thereby increasing the probability of finding higher-quality solutions (Zhang et al., 2015). Following this logic, we generate multiple heatmaps from $p_{\boldsymbol{\theta}}(\mathbf{x}_0|s)$ with different noise data $\mathbf{x}_t$ and then apply the greedy decoding algorithm described above to each heatmap.

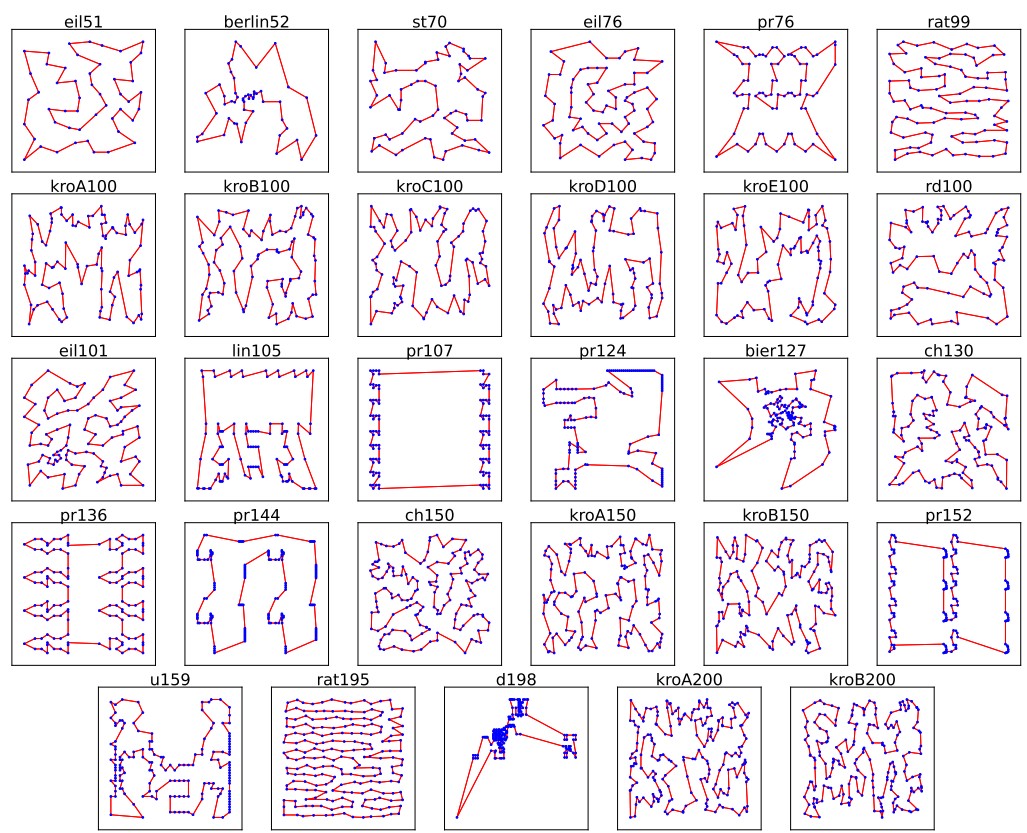

Figure 8: Visualization of DISCO's generated solutions on TSPLIB instances.

**2-opt Strategy**  We also adopt a 2-opt decoding strategy (Andrade et al., 2012) to refine the greedy solutions of TSP tasks. Specifically, we iteratively swap two edges in the current solution to reduce the total length of the tour. We repeat this process until no further improvement can be made. We follow Graikos et al. (2022) and use the Greedy+2-opt strategy as the default.

We conduct DISCO combined with all these strategies on TSP instances to demonstrate its robustness. Following instructions from Böther et al. (2022), we only conduct greedy strategy and sampling strategy on MIS instances to fairly compare the capabilities of different parameterized solvers.

# G   Implementation Details

**Training Setting**  We adopt anisotropic GNNs (Velickovic et al., 2017) as the backbone of our DISCO model. Anisotropic GNNs can produce embeddings for both nodes and edges, which exactly match the CO problems most of which can be formulated as graph problems. Specifically, we input the noisy solution $\mathbf{x}_t$, the node and edge features of $\mathbf{X}_d$, and the time $t$ into the anisotropic GNN with parameter $\boldsymbol{\theta}$, predicting the parameterized residue $\mathbf{x}_{res}^{\boldsymbol{\theta}}$ and noise $\boldsymbol{\epsilon}^{\boldsymbol{\theta}}$ simultaneously with two independent convolutional layers. We use a 12-layer anisotropic GNN with a width of 256 as DIFUSCO (Sun & Yang, 2023) does. We adopt a linear schedule (Huang et al., 2023a) to gradually reduce the noise during the model's generation process.

Following Sun & Yang (2023), we implement sparsification in large-scale graph problems to diminish computational complexity. We constrain each node to maintain only $k$ edges connecting to its closest neighbors. Specifically, we set $k = 100$. We also directly transfer the trained models to the same graphs with different sparsifications without fine-tuning, the generalization results are summarized in Tab. 7. We can observe that DISCO performs consistently stable while the sparsification changes, indicating its robustness.

For TSP-5000 instances, we train DISCO with instances of 64000 and batch size of 12. For TSP-8000/10000, the model is trained using 6,400 instances with a batch size of 4. Align with DIFUSCO, we incorporate curriculum learning approach (Bengio et al., 2009) and begin the training process

Table 7: Evaluation results on different sparsification $k$. The symbol ∘ indicates the sparsification used for training, while the other lines are directly generalized results without fine-tuning.

| Algorithm | Type | $k$ | TSP-5000 | | TSP-8000 | | TSP-10000 | |
| --- | --- | --- | --- | --- | --- | --- | --- | --- |
| | | | Length ↓ | Time ↓ | Length ↓ | Time ↓ | Length ↓ | Time ↓ |
| DISCO | SL+G† | 50 | 52.36 | 5.03m | 66.14 | 13.48m | 73.85 | 24.80m |
| DISCO | SL+G† | 100 | 52.48∘ | 5.72m | 66.11∘ | 14.32m | 73.85∘ | 25.12m |
| DISCO | SL+S | 50 | 52.34 | 7.51m | 66.07 | 22.26m | 73.82 | 40.17m |
| DISCO | SL+S | 100 | 52.44∘ | 9.06m | 66.06∘ | 22.82m | 73.81∘ | 48.77m |

from the TSP-100 checkpoint. For TSP-5000/8000/10000, we label training instances using the LKH-3 heuristic solver (Helsgaun, 2017) with 1000 trials. For the TSP-50 and TSP-100 models used for the checkpoint, we generate 1502000 random instances labeled by Concorde solver, training with batch sizes of 256 and 128 respectively.

For the MIS instances, we use the training split of 49500 examples from SATLIB (Hoos & Stützle, 2000), training with a batch size of 64. For Erdős-Rényi graph sets (Erdős & Rényi, 1960), we use 60000 random instances from the ER-[700-800] variant and train DISCO with a batch size of 16. The training instances are labeled by the KaMIS heuristic solver (Lamm et al., 2016).

**Evaluation Details** We conduct extensive evaluations on both TSP and MIS instances to demonstrate the superiority of our DISCO model. We keep our experimental settings consistent with previous literature (Qiu et al., 2022; Sun & Yang, 2023). For small-scale TSP instances, i.e., TSP-50 and TSP-100, we evaluate on 1280 instances, while for TSP-5000/8000/10000, we use 16 instances. For MIS instances, we evaluate on 500 instances on SATLIB and 128 instances on ER-[700-800].

# H ADDITIONAL RESULTS

## H.1 GENERALIZATION TO UNSEEN PROBLEM DISTRIBUTIONS

We compare the generalization ability of our method to unseen problem distributions with that of the diffusion solver DIFUSCO.

Both methods are trained on uniform distribution and tested on other different distributions for TSP-10000 instances, including a normal distribution $\mathcal{N}(\mu, \sigma^2)$ and a cluster distribution proposed by Bi et al. (2022). For the normal distribution, we set $\mu = 0.5$ and $\sigma^2 = 0.1$ to ensure a distinct difference from the training. This generalization comparison does not include the diffusion solver T2T, due to its use of active search during deployment. The comparisons are summarized in Tab. 8.

Table 8: Evaluation results on different problem distributions.

| Algorithm | Type | Uniform | | Normal | | Cluster | |
| --- | --- | --- | --- | --- | --- | --- | --- |
| | | Length ↓ | Time ↓ | Length ↓ | Time ↓ | Length ↓ | Time ↓ |
| DIFUSCO | SL+G† | 73.99 | 35.38m | 116.76 | 29.27m | 37.84 | 30.25m |
| DISCO | SL+G† | **73.85** | 25.12m | **116.34** | 25.57m | **37.74** | 28.13m |

## H.2 RESOURCE CONSUMPTION COMPARISONS

We compare our algorithm with the state-of-the-art diffusion-based models, DIFUSCO and T2T, in terms of computational resources on TSP-10000 instances as presented in Tab. 9.

Table 9: Comparisons on computational resources.

| Algorithm | Type | Length ↓ | Gap ↓ | Time ↓ | GPU Memory ↓ | GPU hours ↓ |
| --- | --- | --- | --- | --- | --- | --- |
| DIFUSCO | SL+G† | 73.99 | 3.10% | 35.38m | 14G | 0.59 |
| T2T | SL+G† | 73.87 | 2.92% | 91.32m | 71G | 1.52 |
| DISCO | SL+G† | **73.85** | **2.90%** | 25.12m | 14G | 0.42 |

### H.3 GENERALIZATION TO UNSEEN DEGRADED SOLUTIONS

To validate our model's generalization capability on the degraded solution $\mathbf{X}_d$, we conduct experiments using a degraded solution that differs from the training one. The following configurations of degraded solutions are tested:

- Training: We adopt the same $\mathbf{X}_d$ used during the training process, i.e., connecting vertices in the graph sequentially to form a tour.
- Greedy: The nearest unvisited node to the current node is selected as the next step, ensuring the partial solution remains valid. This process is repeated iteratively until a complete path is constructed.
- Far Ins: The farthest insertion algorithm proposed by Cook et al. (2011) is applied to construct the degraded solution. This method iteratively inserts the farthest unvisited node into the tour.
- LKH-3: The degraded solution is generated using the LKH-3 heuristic solver (Helsgaun, 2017) with 1000 trials and 10 runs.

We conduct the comparison on TSP-5000, and the results are presented in Tab. 10. These results demonstrate that our model generalizes effectively across various $\mathbf{X}_d$ configurations.

Table 10: Comparisons on various degraded solution configurations.

| $\mathbf{X}_d$ | TYPE | TRAINING | GREEDY | FAR INS | LKH-3 |
|---|---|---|---|---|---|
| PERFORMANCE | SL+G† | 52.48, 1.04% | 52.47, 1.02% | 52.48, 1.04% | 52.39, 0.87% |

### H.4 CLARIFICATION ON HOW DISCO AVOIDS SUB-OPTIMAL

The graph-splitting approach prevents DISCO from falling into local optima by sampling overlapped subgraphs. This overlap mechanism ensures the same edge can be evaluated from multiple sub-problem views, avoiding local optima caused by a purely single sub-problem view. Then, the generated sub-heatmaps are effectively merged for finally achieving a high-quality solution.

As described in Sec. 4.3, each node is included and evaluated by at least $\omega$ subgraphs simultaneously, which means overlap exists among subgraphs. We design comparative experiments to analyze the effect of this overlap mechanism. As shown in Tab. 11, when there is no overlap between the subgraphs, the model's performance significantly decreases. We also provide a Venn graph illustrating relationships between overlapped subgraphs and an illustration of subgraphs without overlap in Fig. 9.

Table 11: Comparisons of graph splitting with and without overlap.

| ALGORITHM | TYPE | TSP-5000 | TSP-8000 | TSP-10000 |
|---|---|---|---|---|
| DISCO W/O OVERLAP | SL+GS+G | 55.60, 7.05% | 70.42, 7.99% | 82.29, 14.66% |
| DISCO W/ OVERLAP | SL+GS+G | **52.77**, **1.60%** | **66.56**, **2.07%** | **74.45**, **3.73%** |

We also compare different graph merging methods in our multi-modal graph search. In this divide-and-conquer process, each edge may be shared by multiple subgraphs, with corresponding heatmaps sampled for each subgraph. To leverage this information, we employ various merging methods to determine the final value for each edge. Following the notations in Sec. 4.3, we describe each merging method as follows:

- "Min" (or "Max") selects the minimum (or maximum) value for each edge from all corresponding heatmaps, i.e. $\min_{|\mathbf{g}|} \phi(\mathbf{h}_l, i, j)$ (or $\max_{|\mathbf{g}|} \phi(\mathbf{h}_l, i, j)$).
- "Argmin" (or "Argmax") ranks the edge values within each heatmap it belongs to. The value in the final merged heatmap is chosen based on the heatmap where edge $ij$ ranks the lowest (or highest), i.e. $\phi(\mathbf{h}_{\arg\min_{|\mathbf{g}|} \phi(\mathbf{h}_l, i, j)}, i, j)$ (or $\phi(\mathbf{h}_{\arg\max_{|\mathbf{g}|} \phi(\mathbf{h}_l, i, j)}, i, j)$).
- "Mean" calculates the average of all occurrences of each edge across the heatmaps, i.e. $\frac{1}{o_{ij}} \times \sum_{l=1}^{|\mathbf{g}|} \phi(\mathbf{h}_l, i, j)$.

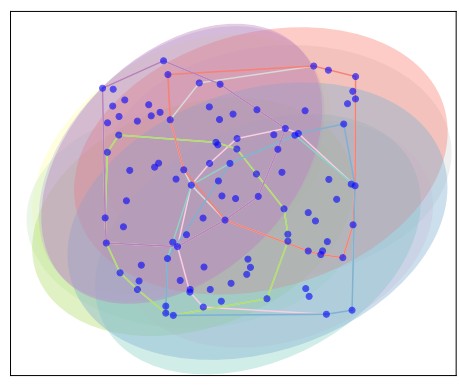 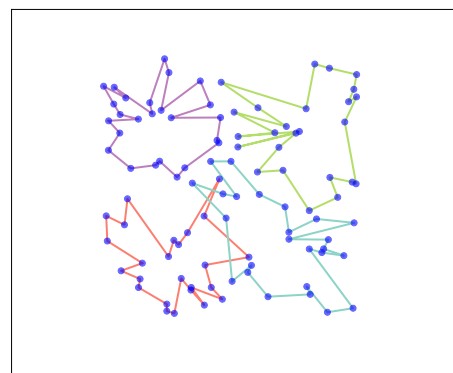

(a) W/ overlap             (b) W/o overlap

Figure 9: Illustrations of subgraphs with overlap (a) and without overlap (b). For clarity, in Figure (a), the subgraph boundary is shown as the convex hull of its vertex set, while in Figure (b), it is represented by connecting its boundary points.

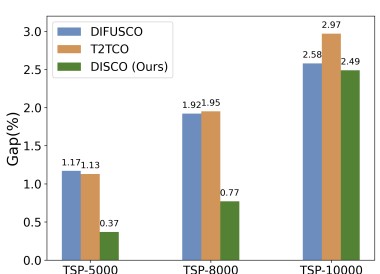

Figure 10: Decoding with the MCTS strategy.

Table 13: Comparisons of three diffusion-based solvers using MCTS as the decoding strategy on TSP-5000, 8000, and 10000. LKH-3 is the baseline for computing the performance gap.

| METHOD | TSP-5000 | | TSP-8000 | | TSP-10000 | |
|---|---|---|---|---|---|---|
| | LENGTH ↓ | GAP ↓ | LENGTH ↓ | GAP ↓ | LENGTH ↓ | GAP ↓ |
| LKH-3 | 51.94* | — | 65.21* | — | 71.77* | — |
| DIFUSCO | 52.55 | 1.17% | 66.46 | 1.92% | 73.62 | 2.58% |
| T2T | 52.66 | 1.13% | 66.48 | 1.95% | 73.90 | 2.97% |
| DISCO (**OURS**) | **52.13** | **0.37%** | **65.71** | **0.77%** | **73.56** | **2.49%** |

We conduct a comparison of various merging methods on TSP-5000 in terms of Length↓ and Gap(%)↓. The results are summarized in Tab. 12.

Table 12: Comparisons on various merging methods.

| MERGING METHOD | TYPE | MIN | MAX | ARGMIN | ARGMAX | MEAN |
|---|---|---|---|---|---|---|
| PERFORMANCE | SL+G† | 79.08, 52.25% | 53.05, 2.14% | 79.15, 52.39% | 52.94, 1.93% | **52.77, 1.60%** |

### H.5 COMPARISONS USING MCTS AS DECODING STRATEGY

We compare our method with recent diffusion solvers DIFUSCO (Sun & Yang, 2023) and T2T (Li et al., 2024), taking MCTS as the decoding strategy. The results are presented in Tab. 13. A bar chart illustrating the performance discrepancy among each method is also provided in Fig. 10. Our method outperforms the others across all three scales.

### H.6 RESULTS ON TSP-500 AND TSP-1000

We provide more comprehensive comparisons on TSP-500 and TSP-1000. We extensively compare DISCO with various baselines, including exact solvers, heuristic solvers, and recent non-diffusion-based learning methods. For exact solvers, our comparisons include Concorde (Applegate et al., 2006) and Gurobi (LLC Gurobi Optimization, 2018). Regarding heuristic solvers, we evaluate against LKH-3 (Helsgaun, 2017) and 2-opt (Croes, 1958). In terms of learning-based methods, we compare with recent non-diffusion neural solvers including AM (Kool et al., 2019), GCN (Joshi et al., 2019), ELG-POMO (Gao et al., 2023), BQ-NCO (Drakulic et al., 2024), and GLOP (Ye et al., 2024b). The results are shown in Tab. 14

Table 14: Comparisons on TSP-500 and TSP-1000. G denotes Greedy decoding. The symbol * indicates the baseline for computing the performance gap. The symbol † denotes that the diffusion model samples once.

| ALGORITHM | TYPE | TSP-500 LENGTH ↓ | GAP ↓ | TIME ↓ | TSP-1000 LENGTH ↓ | GAP ↓ | TIME ↓ |
|---|---|---|---|---|---|---|---|
| CONCORDE | EXACT | 16.55* | — | 37.66m | 23.12* | — | 6.65h |
| GUROBI | EXACT | 16.55 | 0.00% | 45.63h | N/A | N/A | N/A |
| LKH-3 (DEFAULT) | HEURISTICS | 16.55 | 0.00% | 46.28m | 23.12 | 0.00% | 2.57h |
| RAW 2-OPT | HEURISTICS | 17.99 | 8.68% | 0.33m | 25.24 | 9.16% | 1.08m |
| AM | RL+G | 19.99 | 20.79% | 1.08m | 31.12 | 34.60% | 1.15m |
| GCN | SL+G | 29.72 | 79.61% | 6.67m | 48.62 | 110.29% | 28.52m |
| BQ-NCO | RL+G | 16.97 | 2.54% | 1.56m | 23.92 | 3.48% | 11.03m |
| ELG-POMO | RL+G | 17.66 | 6.71% | 3.88m | 25.65 | 10.94% | 22.87m |
| GLOP | RL+G | 16.91 | 1.99% | 1.50m | 23.84 | 3.11% | 3.00m |
| DISCO (**OURS**) | SL+G† | **16.86** | **1.87%** | 0.25m | **23.65** | **2.29%** | 1.12m |

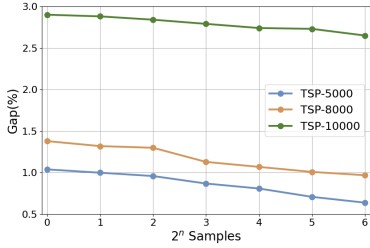

Figure 11: Solution quality improves as the number of samples increases.

Table 15: Comparisons of performances in terms of number of samples. LKH-3 is the baseline for computing the performance gap.

| SAMPLE | TSP-5000 LENGTH ↓ | GAP ↓ | TSP-8000 LENGTH ↓ | GAP ↓ | TSP-10000 LENGTH ↓ | GAP ↓ |
|---|---|---|---|---|---|---|
| LKH-3 | 51.94* | — | 65.21* | — | 71.77* | — |
| 1 | 52.48 | 1.04% | 66.11 | 1.38% | 73.85 | 2.90% |
| 2 | 52.66 | 1.13% | 66.48 | 1.95% | 73.90 | 2.97% |
| 4 | 52.44 | 0.96% | 66.06 | 1.30% | 73.81 | 2.84% |
| 8 | 52.39 | 0.87% | 65.95 | 1.13% | 73.77 | 2.79% |
| 16 | 52.36 | 0.81% | 65.91 | 1.07% | 73.74 | 2.74% |
| 32 | 52.31 | 0.71% | 65.87 | 1.01% | 73.73 | 2.73% |
| 64 | 52.27 | 0.64% | 65.84 | 0.97% | 73.67 | 2.65% |

## H.7 ENHANCING SOLUTION QUALITY THROUGH MULTI-MODAL PROPERTIES

We conduct a direct experiment to demonstrate the impact of the multi-modal property on improving solution quality. We vary the number of noises for sampling solutions. We present a line chart in Fig. 11 with the number of samples as the x-axis and Gap(%)↓ as the y-axis to visually demonstrate how the multi-modal property enhances model performance. The detailed comparison results are outlined in Tab. 15.

