# OpenReview forum: "DISCO: Efficient Diffusion Solver for Large-Scale Combinatorial Optimization Problems"
_ICLR.cc/2025/Conference — Submitted to ICLR 2025_

### Official Review · Reviewer_GPzi · 2024-11-01

**Soundness:** 3
**Presentation:** 3
**Contribution:** 2
**Rating:** 5
**Confidence:** 4

**Summary:**

Summary
This paper introduces DISCO, a diffusion-based solver for combinatorial optimization  problems, using denoising diffusion probabilistic models (DDPM) with added solution residues to restrict search space. The forward diffusion process maps ground-truth solutions to a mixture of noise and degraded solutions. The authors also propose an accelerated sampling approach with fewer steps based on decoupled diffusion models (DDM). Experimental results on TSP and MIS demonstrate that the proposed method achieves the improved solution quality and a faster sampling.

**Strengths:**

Strengths
The paper is well-structured and presents ideas in a clear, logical manner, and easy to read.
The proposed method demonstrates improvements on TSP and MIS.

**Weaknesses:**

Weaknesses
Incremental Innovation: While DISCO leverages DDPMs for CO problems with the inclusion of solution residues, the approach appears incremental. Prior work on CO tasks has explored conditioning on initial solutions, which limits the novelty here.
The technical methodology seems to follow the conventional DDPM structure closely. A clearer breakdown of the challenges specific to CO and the innovations made by DISCO would strengthen the contribution.
The proposed fast sampling process appears similar to existing DDM techniques, with limited innovation beyond adding residues.
What is the performance (solution quality and sampling time) of the DIFUSCO baseline with DDM as sampler?
The performance improvement over DIFUSCO is marginal.
The claim regarding DISCO’s multi-modal property requires further justification. How this property enhances solution quality or contributes to CO performance is unclear without additional evidence or explanation.

**Questions:**

NA

---

### Official Review · Reviewer_AWgu · 2024-11-01

**Soundness:** 3
**Presentation:** 3
**Contribution:** 2
**Rating:** 6
**Confidence:** 4

**Summary:**

This paper presents a new approach, DISCO, designed to efficiently solve large-scale combinatorial optimization  problems using diffusion models. DISCO addresses two primary challenges in current diffusion solvers: sampling from extensive NP-complete solution spaces and the time-intensive denoising processes. By incorporating solution residues, DISCO focuses on meaningful solution domains, maintaining the multi-modal properties of CO problems while reducing computational overhead. The model also employs an analytically solvable denoising process that significantly cuts down inference time. DISCO demonstrates notable performance improvements on tasks such as the TSP and MIS. The method generalizes well to unseen problem scales through a divide-and-conquer approach, sometimes surpassing models trained specifically for those scales.

**Strengths:**

1.	The idea of using a feasible solution as the residue constrain to guide the generation process sounds novel and reasonable.
2.	Experimental results show that DISCO is both efficient and effective in solving large scale TSP problems.

**Weaknesses:**

1.	The novelty is limited. Solution residue is from [1] and multi-model is from [2]. Can you summarize the novelty of the proposed method?
2.	It is unclear why the residue term can lead to better solutions. Figure 1 is just an explanation of the intuition. More supporting analysis and evidence are needed.

[1] Liu, Jiawei, et al. "Residual denoising diffusion models." Proceedings of the IEEE/CVF Conference on Computer Vision and Pattern Recognition. 2024.
[2] Fu, Zhang-Hua, Kai-Bin Qiu, and Hongyuan Zha. "Generalize a small pre-trained model to arbitrarily large tsp instances." Proceedings of the AAAI conference on artificial intelligence. Vol. 35. No. 8. 2021.

**Questions:**

1.	“X_d can be obtained by connecting vertices in the graph in a sequential order to form a tour.” I think X_d may have a huge impact on the solutions. How to ensure that X_d always guides the correct search direction? In training and testing, X_d is different. How can you ensure the effectiveness of trained model on test data?
2.	MCTS shows better performance than sampling in existing works and Table 3. The authors claim that “Xia et al. (2024) highlight that the MCTS strategy (Fu et al., 2021) heavily relies on TSP-specific heuristics, and is less suited to other problem types.”, but they are still solving TSP in Table 1. It is not reasonable to choose sampling rather than MCTS.
3.	Why the authors choose TSP-5000, 8000, and 10000? Does the proposed method still work on TSP 100, 500, 1000?
4.	In Table 3, why ATT-GCN performs very bad on TSP-5000, 8000 but performs much better and faster on TSP-10000?

---

### Official Review · Reviewer_Xho3 · 2024-11-04

**Soundness:** 3
**Presentation:** 3
**Contribution:** 2
**Rating:** 6
**Confidence:** 4

**Summary:**

The paper presents DISCO, a novel algorithm developed to efficiently address large-scale combinatorial optimization (CO) problems. DISCO effectively manages the multi-modal complexity of CO landscapes, enabling swift solution generation and delivering high-quality results with notably fewer computational steps. Tested on extensive benchmarks, including large-scale Traveling Salesman Problems (TSP) and Maximal Independent Set (MIS), DISCO demonstrates state-of-the-art performance in both solution quality and inference speed.

**Strengths:**

1. DISCO achieves state-of-the-art results on large-scale TSP-10000 instances and challenging MIS benchmarks, demonstrating superior performance both in terms of solution quality and speed.

2. The proposed divide-and-conquer strategy effectively generalizes DISCO to solve large-scale problem instances, highlighting the method’s scalability and versatility.

3. Beyond the previous results, the paper extends experiment results on aspects of 1) [which I think is the most significant challenge for the supervised learning framework] illustration of the generalization ability of DISCO 2) comparison of the time consumption and computational workload of DISCO 3) add more powerful baseline T2T[1] as comparison. These empirical results further validates the solidness of the paper.

[1] T2T: From Distribution Learning in Training to Gradient Search in Testing for Combinatorial Optimization, NeurIPS 2023.

**Weaknesses:**

The graph search might lead to exponential growth in trial variance.The scalability might still remain an issue. The authors have claimed the issue in limitation and leave it as future research oppurtunity.

**Questions:**

From my perspective, the experiments are intensive enough to illustrate the effectiveness of the proposed method, and thus I have no more questions.

---

### Official Review · Reviewer_PM6T · 2024-11-04

**Soundness:** 3
**Presentation:** 3
**Contribution:** 2
**Rating:** 6
**Confidence:** 3

**Summary:**

The paper introduces DISCO, a diffusion-based solver optimized for large-scale combinatorial optimization (CO) problems, such as the Traveling Salesman Problem (TSP) and Maximal Independent Set (MIS). Unlike traditional diffusion models, DISCO focuses on constraining the solution space to enhance quality and employs an analytically solvable denoising approach, which speeds up the process. Its twofold strategy—guided sampling and efficient denoising—achieves significantly faster inference times while maintaining high solution accuracy. DISCO's multi-modal search approach also allows it to generalize effectively to unseen problem scales.

**Strengths:**

1.	DISCO significantly reduces inference times by introducing an analytically solvable denoising process and constraining the sampling space.
2.	The paper is well-written.

**Weaknesses:**

1.	The multi-modal graph search is presented as a crucial component of DISCO, with the paper asserting that its multi-modal output helps prevent sub-optimal solutions. However, the experiments do not thoroughly analyze the contribution of this module. It would be beneficial for the authors to include performance metrics with and without this module and detail the computational overhead incurred when it is enabled.
2.	The model's approach involves initially splitting the graph to find solutions, yet the problems tested (TSP, MIS) are inherently global, where localizing could risk sub-optimal outcomes. Could the authors clarify how this graph-splitting approach supports the model’s effectiveness in avoiding sub-optimal solutions for these global problems

**Questions:**

Please refer to the weaknesses.

---

> ### Comment · Reviewer_PM6T · 2024-11-22
>
> Thank you for your respond! My concerns have been addressed, and I will raise my score.

---

> ### Author Response · Authors · 2024-11-22
>
> Thank you very much for recognizing the effectiveness of DISCO's multi-modal graph search and for raising your score! Your valuable insights have been instrumental in making our paper more comprehensive. We have incorporated your constructive comments into the revision. Thanks once more for your time and effort in reviewing our paper!

---

### Comment · Reviewer_GPzi · 2024-11-26

Thank you to the authors for their detailed responses to my concerns. After carefully reviewing the rebuttal and considering the comments from other reviewers, I have decided to maintain my score. While the work is well-executed, the use of conditional guidance, which is central to the proposed method, is already widely adopted in diffusion models for various tasks. As a result, the novelty of the contribution is somewhat limited.

---

> ### Author Response · Authors · 2024-11-27
>
> Dear Reviewer GPzi:
>
> We respectfully disagree with your opinion. DISCO is the first residue-guided diffusion solver specifically designed to address large-scale combinatorial optimization (CO) problems. It creatively introduces residue-restricted search spaces into the solving process for CO problems and uses an analytical denoising process to accelerate solution generation, enabling higher-quality solutions for large-scale CO problems in a shorter time. This is particularly crucial in solving CO problems given the exponential expansion of the solution space as the problem scale grows.
>
> DISCO has already demonstrated its effectiveness on both edge-based TSP and node-based MIS problems, which stand out as two foundations of CO regarding edge and node decision problems. This highlights DISCO's potential as an efficient and general-purpose solver for the broader CO domain. If the reviewer knows of any similar residue-guided diffusion solvers with open-source implementations, we are very glad to provide performance comparisons and discuss the differences.

---

### Author Response · Authors · 2024-12-04
**Summary of the Discussion (2/2)**

Based on the discussion with reviews, we also present a brief summary of our paper as follows:

- **Observation**: Existing diffusion-based CO solvers overlook the inefficient solution sampling from enormous NPC solution space and the slow reverse process of diffusion models, limiting their applicability to large-scale real-world problems.
- **Solution**: DISCO addresses these issues by introducing residue-constrained denoising to produce high-quality solutions with fewer steps. It also incorporates a multi-modal graph search mechanism to generalize to unseen problem scales without retraining.
- **Results**: DISCO significantly reduces inference times while maintaining high solution accuracy, as demonstrated in comparisons on large-scale TSP-5000/8000/10000 against mainstream baselines. Furthermore, DISCO’s multi-modal graph search enables effective generalization to unseen problem scales, even surpassing models trained specifically for those scales.
- **Highlights**: Designed to solve large-scale CO problems, our work has the following highlights:
    - **Residue-constrained solution generation**: Residue term can effectively constrain the denoising process, contributing to higher solution quality.
    - **Multi-modal graph search**: The multi-modal property of the diffusion model enables diverse solution generation, enhancing performance during graph search.

Thanks again for your efforts in the reviewing and discussion. We appreciate all the valuable feedback that helped us to improve our submission.

Sincerely

Authors of Submission 6285

---

### Author Response · Authors · 2024-12-04
**Summary of the Discussion (1/2)**

Dear Chairs and Reviewers,

Hope this message finds you well.

As the discussion period concludes, we present a brief summary of our discussion with the reviewers as an overview for reference. First, we sincerely thank all reviewers for their insightful comments and constructive suggestions. We are encouraged by the positive recognition of our work, including:

- **Reviewer PM6T**: Acknowledged that DISCO significantly reduces inference times while maintaining high solution accuracy. Praised the clarity of the paper and noted that the multi-modal graph search approach generalizes effectively to unseen problem scales.
- **Reviewer Xho3**: Highlighted DISCO's superior performance in both solution quality and speed, characterizing it as novel, scalable, and versatile. The empirical results further validated the robustness of DISCO.
- **Reviewer AWgu**: Recognized the innovative use of residue constraints to guide the generation process, regarding it as novel and reasonable. Emphasized DISCO’s efficiency and effectiveness in solving large-scale TSP problems, along with its strong generalization capabilities to unseen scales.
- **Reviewer GPzi**: Appreciated the clear structure, well-execution, and readability of the paper, pointing out that DISCO achieves improved solutions with faster sampling.

We have carefully read all the comments and responded to them in detail. All raised concerns have been addressed in our revised manuscript, with the corresponding changes colored in red.

---

We summarize the main concerns of the reviewers with the corresponding response as follows:

- **Novelty of Our Method**
    - DISCO’s novelty lies in adopting residue to constrain solution space to a more meaningful region and leveraging multi-modal property in graph search to avoid local optima. Our theoretical analysis, superior performance on large-scale TSP and MIS problems, along with step-wise denoising comparison and generalization experiments on degraded solutions $\mathbf{X}_d$ demonstrate the effectiveness and versatility of our method.
    - DISCO‘s advantages on both edge-based TSP and node-based MIS problems, which stand out as two foundations of CO regarding edge and node decision problems, highlight its potential as an efficient and general-purpose solver for the broader CO domain. To our knowledge, DISCO is the first residue-guided diffusion solver specifically designed for large-scale combinatorial optimization (CO) problems.
- **Contribution of Multi-modal Graph Search in Avoiding Local Optima**
    - We provide experimental results comparing our algorithm with and without the multi-modal graph search module, focusing on both performance (Length↓ and Gap↓) and computational resource usage (GPU memory and GPU hours). Comparisons show that performance improves as sub-heatmap diversity increases, corroborating the critical role of the multi-modal property in avoiding local optima.
    - We further demonstrate DISCO's overlapping mechanism and effective merging method also help avoid local optima. Experiments show that without overlap, the performance significantly decreases.
- **Justifications for DISCO's Multi-modal Property Enhancing Solution Quality**
    - By varying the number of noise samples used in the diffusion model's sampling process, we demonstrate that increasing noise samples enhances solution diversity. This effectively leverages the model’s multi-modal property, increasing the likelihood of identifying higher-quality solutions.
- **Additional Experiments**
    - We compare DISCO against DIFUSCO and T2T with MCTS decoding, showing that DISCO outperforms these methods on all tested scales.
    - Further experiments on smaller-scale TSPs and real-world TSPLIB instances reinforce DISCO’s robustness across diverse problem settings.

---

---

### Meta-Review · Area_Chair_E4hk · 2024-12-20

**Metareview:**

(a) Summarize the scientific claims and findings: The paper claims DISCO achieves faster inference times and higher accuracy on large-scale combinatorial optimization problems like TSP and MIS by introducing residue-constrained denoising and a multi-modal graph search, but reviewers have raised concerns about the novelty and significance of these contributions.

(b) What are the strengths of the paper? The paper is well-written and presents a comprehensive evaluation of DISCO on large-scale TSP and MIS problems, demonstrating its efficiency and generalization ability.

(c) What are the weaknesses of the paper? Despite the authors' claims, reviewers find the novelty and significance of the contributions limited, particularly the use of conditional guidance and the multi-modal graph search, which are already adopted in existing diffusion models.

(d) Provide the most important reasons for your decision to reject. While the paper presents a well-executed approach, the limited novelty and marginal improvements over existing methods do not meet the bar for acceptance at this venue, where significant contributions to the field are expected.

**Additional Comments On Reviewer Discussion:**

The reviewers all communicated with the authors during the discussion phase, but some issues remained, and there was no strong enthusiasm in favor of this paper compared to other submissions.

---

### Decision · Program_Chairs · 2025-01-22

Reject